# Report

# IFT proteins interact with HSET to promote supernumerary centrosome clustering in mitosis

Benjamin Vitre[1,2,*] (iD), Nicolas Taulet[1,2], Audrey Guesdon[1,2], Audrey Douanier[1,2], Aurelie Dosdane[1,2], Melanie Cisneros[1,2], Justine Maurin[1,2], Sabrina Hettinger[1,2], Christelle Anguille[1,2], Michael Taschner[3,†], Esben Lorentzen[3] (iD) & Benedicte Delaval[1,2,**] (iD)

## Abstract

Centrosome amplification is a hallmark of cancer, and centrosome clustering is essential for cancer cell survival. The mitotic kinesin HSET is an essential contributor to this process. Recent studies have highlighted novel functions for intraflagellar transport (IFT) proteins in regulating motors and mitotic processes. Here, using siRNA knock-down of various IFT proteins or AID-inducible degradation of endogenous IFT88 in combination with small-molecule inhibition of HSET, we show that IFT proteins together with HSET are required for efficient centrosome clustering. We identify a direct interaction between the kinesin HSET and IFT proteins, and we define how IFT proteins contribute to clustering dynamics during mitosis using high-resolution live imaging of centrosomes. Finally, we demonstrate the requirement of IFT88 for efficient centrosome clustering in a variety of cancer cell lines naturally harboring supernumerary centrosomes and its importance for cancer cell proliferation. Overall, our data unravel a novel role for the IFT machinery in centrosome clustering during mitosis in cells harboring supernumerary centrosomes.

**Keywords** cancer; centrosome clustering; HSET/KifC1; IFT proteins; mitosis
**Subject Categories** Cell Adhesion, Polarity & Cytoskeleton; Cell Cycle

## Introduction

The centrosome is a dynamic organelle that has essential functions in both cycling and non-cycling cells. In non-cycling cells, the centrosome/basal body contributes to cell polarization, motility, and primary cilia formation [1]. In dividing cells, centrosomes contribute to the assembly of a functional bipolar mitotic spindle [2,3]. Having only two centrosomes per mitotic cell is essential

for accurate chromosome segregation. Indeed, the presence of supernumerary centrosomes, known as centrosome amplification, promotes multipolar spindle formation that can drive chromosome segregation errors and aneuploidy [4–8] resulting in tumorigenesis initiation and/or acceleration [9–12]. Despite those pro-tumorigenic effects, supernumerary centrosomes and multipolar spindles that result in multipolar anaphases are, most of the time, detrimental for cell proliferation and survival [4,7,8,13–16]. However, cancer cells that often present high levels of centrosome amplification are capable of clustering those centrosomes to favor the formation of (pseudo)bipolar mitotic spindles. This process allows cancer cell proliferation and survival [4,13,17]. Multiple studies have high-lighted the importance of the mitotic checkpoint and the actin/microtubule networks, together with their associated proteins and motors, for efficient centrosome clustering [9,13,17,18]. Among those microtubule-associated motors, the minus-end-directed kinesin HSET/KIFC1 is of particular interest, since it is essential for supernumerary centrosome clustering but not for the division of cells with two centrosomes [9,13,19–21].

IFT proteins form polarized cargo transport complexes that function in both non-dividing and dividing cells in association with microtubules and motors. They were originally described as large molecular complexes transported by molecular motors along the length of cilia and flagella, and they are essential for their assembly and function [22–24]. IFT proteins are classified into two complexes: IFT complex A and IFT complex B. Within the IFT B complex, IFT88/70/52/46 subunits are part of the IFT-B core subcomplex (IFT-B1 [25,26]).

In addition to their role in cilia, multiple studies recently showed that IFT proteins also function outside of the ciliary compartment, in a variety of cellular processes. Indeed, IFT proteins are important for immunological synapse organization in non-ciliated lymphocytes [27] and for interphase microtubule cytoskeleton organization and dynamics [28–30]. IFT proteins also contribute to multiple aspects of cell division. In prometaphase, IFT88 contributes to NuMA accumulation at k-fiber minus ends, favoring their incorporation into the

1   CRBM, University of Montpellier, CNRS, Montpellier, France
2   Centrosome, Cilia and Pathologies Lab, Montpellier, France
3   Department of Molecular Biology and Genetics, Aarhus University, Aarhus, Denmark
    *Corresponding author. Tel: +33 4343 59531; E-mail: benjamin.vitre@crbm.cnrs.fr
    **Corresponding author. Tel: +33 4353 59530; E-mail: benedicte.delaval@crbm.cnrs.fr
    †Present address: University of Lausanne, Department of Fundamental Microbiology, Lausanne, Switzerland

mitotic spindle and contributing to proper chromosome alignment [31]. In cytokinesis, an IFT-B subcomplex made of IFT88/70/52/46 directly interacts with the kinesin MKLP2, regulates the relocalization of Aurora B to the spindle midzone, and subsequently shapes central spindle microtubule architecture [32]. Finally, IFT88 in complex with cytoplasmic dynein 1 contributes to the relocalization of peripheral microtubule clusters toward the spindle poles to ensure the proper formation of astral microtubule arrays and correct spindle orientation [33].

Similarities between this latter mechanism and the process of centrosome clustering, *per se*, led us to hypothesize that IFT proteins could contribute to centrosome clustering in cells harboring supernumerary centrosomes. We also hypothesized that this activity of IFT proteins could be mediated by their interaction with microtubule-associated motors involved in this process, such as dynein or the kinesin HSET/KIFC1.

Here, we establish that proteins of the IFT-B subcomplex are required for centrosome clustering. Using siRNA knock-down of various IFT proteins or auxin-inducible degradation of endogenous IFT88 in combination with small-molecule inhibitors, we show the functional contribution of IFT proteins together with both HSET and dynein to the process of centrosome clustering. Moreover, we identify a direct interaction, *in vitro*, between the kinesin HSET and IFT-B subcomplex proteins. Taking advantage of high-resolution imaging of centrosome dynamics, we also show that IFT52 knock-down decreases centrosome clustering dynamics, both in late G2 and in mitosis, to the same extent as HSET small-molecule inhibition. This result suggests a role for IFT proteins in centrosome clustering through the modulation of mitotic motors. Finally, we demonstrate the importance of IFT88 and IFT52 for efficient centrosome clustering in cancer cell lines naturally harboring supernumerary centrosomes and the importance of IFT88 for cancer cell proliferation. Overall, our data unravel a novel role for the IFT machinery in centrosome clustering during mitosis in cells harboring supernumerary centrosomes.

## Results and Discussion

### IFT88 and IFT52 are required for centrosome clustering

Previous studies have shown that members of the IFT-B subcomplex such as IFT88 could interact with and/or modulate motors during mitosis including dynein 1 [33] or MKLP2 [32]. In order to test whether IFT proteins are involved in centrosome clustering, we depleted various IFT proteins from RPE-1 cells with supernumerary centrosomes. In these cells, supernumerary centrosomes were generated by overexpressing the kinase Plk4 under the control of a doxycycline-inducible system (Fig EV1A; [34]). We observed that both IFT52 and IFT88 decorate the mitotic spindle poles in the cells (Figs 1A and EV1B). To assess the impact of IFT protein depletion on centrosome clustering, we quantified the occurrence of multipolar anaphases using live-cell analysis (Fig 1B and C, Movies EV1 and EV2). We found that while siRNA depletion of IFT27 did not affect the proportion of cells undergoing multipolar anaphases, depletion of either IFT88 or IFT52 induced a twofold increase (10.8% in siCT versus 21.6 and 22.8% for siIFT88 and siIFT52, respectively) in the proportion of cells undergoing multipolar

anaphases compared to control siRNA-transfected cells (Fig 1D and E). Importantly, the defects observed upon depletion of human IFT52 were rescued by re-expressing mouse IFT52 in IFT52-depleted cells, validating the specificity of the siRNA (Figs 1E and EV1C). Similar results were obtained in a different RPE-1 cell line overexpressing Plk4 as described above and in which DNA content was visualized using H2B-EGFP instead of SiR-DNA staining to label the DNA (Fig EV1D and E). To assess whether IFT52 and IFT88, which belong to the same subcomplex of IFT-B [25], function together in the process of centrosome clustering, we performed the co-depletion of IFT88 and IFT52 (Fig EV1F and G). The level of multipolar anaphases observed after co-depletion was not significantly different from single IFT protein depletion. This result confirms that IFT88 and IFT52 are most likely acting together in the process of centrosome clustering. In order to control that IFT protein depletion does not affect the mitotic spindle or centrosome integrity, we analyzed immunofluorescence images of RPE-1 cells. No significant global defects in spindle organization were observed upon IFT protein depletion in cells with or without centrosome amplification (Fig EV1H). Indeed, IFT protein depletion by itself was not sufficient to induce multipolar spindle formation in mitotic cells without centrosome amplification (Fig EV1I). Moreover, IFT protein depletion did not weaken centriolar cohesion within the centrosomes. Indeed, the proportion of spindle pole with less than two centrioles remained unchanged (Fig EV1J and K). Finally, defects in astral microtubules described in other cell lines, such as LLC-PK1 [33], were also observed in RPE-1 cells with no centrosome amplification (Fig EV1H, upper panels).

Then, to achieve a complete depletion of IFT proteins in cells using an alternative approach, we targeted both endogenous alleles of IFT88 in DLD-1 cells with an auxin-inducible degron (AID) that triggers rapid depletion of the tagged protein upon auxin addition [35,36]. Complete depletion of IFT88 was achieved following auxin addition to the culture medium (Fig 1F and G). Centrosome amplification was achieved by doxycycline-induced Plk4 kinase overexpression (Figs 1H and EV1A and EV2A). Complete depletion of IFT88 induced a 30% increase in the proportion of multipolar anaphases in DLD-1 cells with centrosome amplification (28.2% in CT versus 42.9% in auxin condition, Fig 1I and J; Movies EV3 and EV4). Similar results were obtained in DLD-1 cells and tetraploid HCT-116 cells harboring supernumerary centrosomes in which IFT88 was depleted using siRNA (Fig EV2B–E). This further validated the importance of IFT proteins for efficient clustering in a variety of cell lines. Of note, as described for RPE-1 cells, acute IFT88 depletion upon auxin addition did not affect global spindle architecture or centriolar cohesion and did not trigger multipolar mitosis in mitotic cells with no centrosome amplification (Fig EV2A, F and G).

Altogether, these results indicate that IFT52 and IFT88 are necessary for efficient clustering of supernumerary centrosomes in a variety of cell types.

### IFT52 and IFT88 act in association with dynein and HSET for efficient centrosome clustering

Dynein and the kinesin HSET/KIFC1 are two of the main mitotic motors involved in centrosome clustering [13,17]. Moreover, recent studies showed that IFT proteins function together with mitotic motors [32,33] or can even control kinesin activity *in vitro* [37]. To

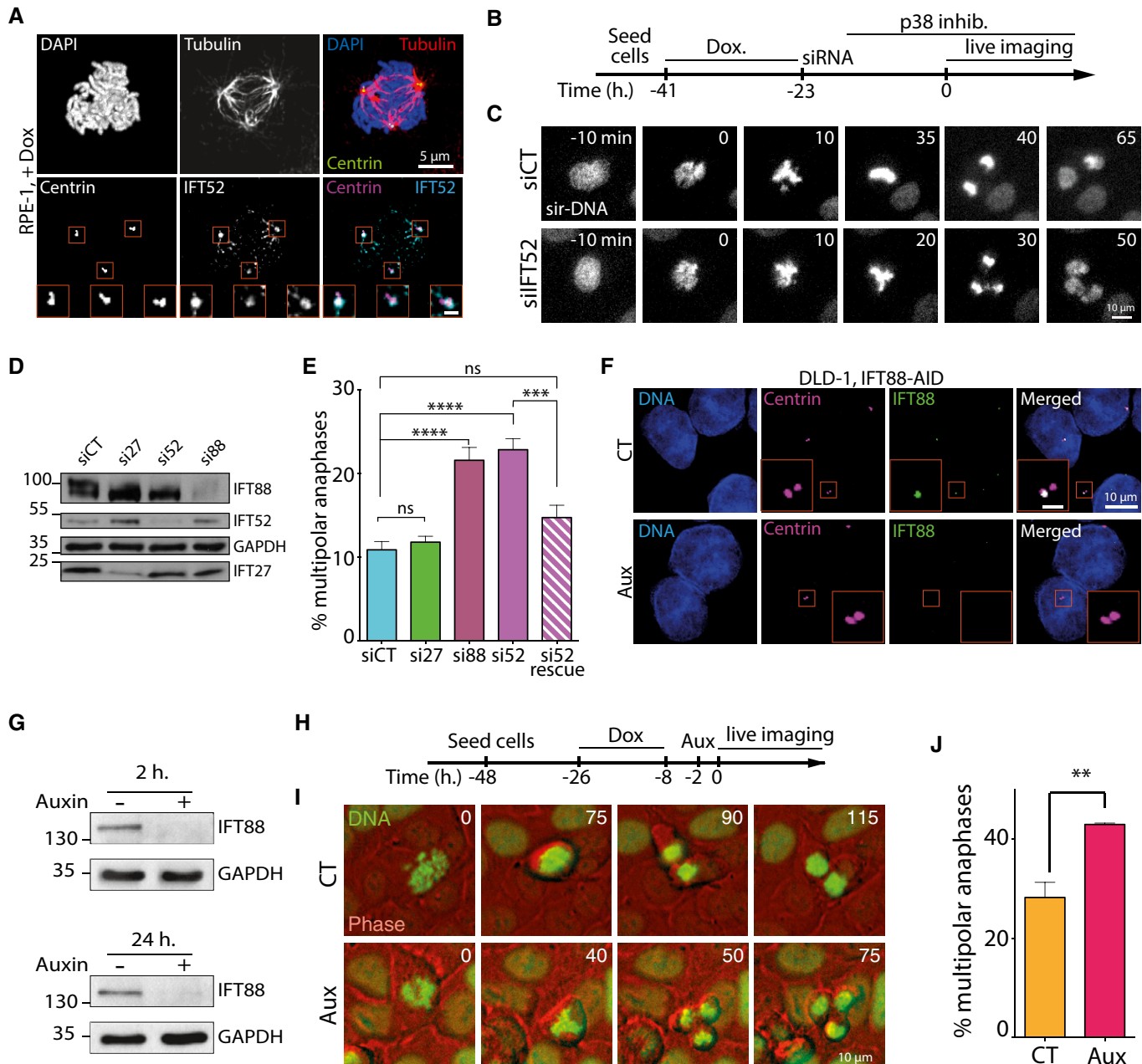

**Figure 1. IFT proteins contribute to centrosome clustering.**

A  Immunofluorescence images of a mitotic RPE-1 cell, with centrosome amplification, showing IFT52 localization. Magnified boxes, spindle poles. Scale bar in magnified box, 1 μm.

B  Schematic of experimental timeline for RPE-1 cells used in panels (B–E).

C  Representative still images extracted from a movie of RPE-1 cells, labeled with SiR-DNA. siRNA treatments are indicated.

D  Immunoblots showing IFT proteins depletion 30 h after siRNA treatment in RPE-1 cells. GAPDH, loading control.

E  Quantification of the percentage of multipolar anaphases following the indicated siRNA treatment. Mean ± SEM from a minimum of four independent experiments. ****$P$ < 0.0001; ***$P$ < 0.001; ns: non-significant; unpaired $t$-test. For each condition, a minimum of 600 cells were quantified.

F  Immunofluorescence images of DLD1 cells expressing endogenous IFT88-AID-YFP. IFT88 was labeled using an anti-GFP antibody to show its depletion from centrioles after auxin treatment. Scale bar in magnified box, 2 μm.

G  Immunoblots showing endogenous IFT88-AID-YFP depletion in DLD-1 cells following various times of auxin treatment.

H  Schematic of experimental timeline for DLD-1 cells used in panels (H–J).

I  Representative still images extracted from a movie of DLD-1 cells with supernumerary centrosomes, expressing H2B-EGFP and treated with or without auxin (Aux) in order to degrade endogenous IFT88 tagged with AID.

J  Quantification of the percentage of multipolar anaphases in DLD-1 IFT88-AID following the indicated treatments. CT: control condition; Aux: auxin treatment. Mean ± SEM from three independent experiments. **$P$ < 0.01; unpaired $t$-test. For each condition, a minimum of 373 cells were quantified.

Source data are available online for this figure.

test whether those motors function together with IFT52 to mediate centrosome clustering, we used small-molecule inhibitors of dynein or HSET in combination with the depletion of IFT52. As expected, the use of ciliobrevin D, a small-molecule inhibitor of the AAA+ ATPase activity of dynein [38], at 20 μM, resulted in a twofold increase in the proportion of multipolar anaphases in RPE-1 cells with supernumerary centrosomes (11.8% in DMSO control condition versus 22.3% in ciliobrevin-treated condition). This result was similar to the effects of IFT52 depletion in those cells (22.9%, Fig 2A–C). Importantly, combining IFT52 depletion and dynein inhibition did not increase the proportion of multipolar anaphases, compared to single perturbation. This result suggested that IFT88 and dynein could function together in centrosome clustering. Similarly, the use of 25 μM of the allosteric inhibitor of the kinesin HSET, CW069 [39], resulted in an increased proportion of multipolar anaphases similar to the increase observed with IFT52 depletion (21.8% versus 22.9%, respectively, Fig 2A). Importantly, combining HSET inhibition and IFT52 depletion did not lead to cumulative defects, suggesting that IFT52 and HSET could function together to facilitate centrosome clustering in RPE-1 cells with supernumerary centrosomes. To confirm this observation in RPE-1 cells using an alternative approach, we used a combination of siRNA targeting both IFT52 and HSET (Fig EV2H and I). As expected, the strong depletion of HSET achieved using siRNA led to a sixfold increase compared to the control condition. Importantly, co-depletion of IFT52 and HSET did not significantly change the level of multipolar anaphases compared to the single depletion of HSET (Fig EV2I). This confirmed that IFT52 and HSET could function together in centrosome clustering.

Taking advantage of the auxin-inducible degradation, we also tested whether IFT88 could function in association with motors to allow centrosome clustering. To exclude that the mitotic defects observed could result from a prolonged depletion of IFT proteins in interphase, we arrested cells in mitosis at the metaphase/anaphase transition and induced the degradation of IFT88-AID with auxin. To arrest cells at the metaphase/anaphase transition, we used two small molecules, APCIN and proTAME, to synergistically block APC/C-Cdc20 interaction (Fig 2B–E) [40]. Using immunofluorescence experiments to quantify mitotic cells with unclustered centrosomes, we found that auxin treatment triggered a 30% increase in the proportion of cells with unclustered centrosomes, compared to control-treated cells (38.8% in CT versus 59.3% in auxin, Fig 2F). As described above in RPE-1 cells, treatments with ciliobrevin D (50 μM) or CW069 (150 μM) led to an increase in DLD-1 cells with unclustered centrosomes similar to the increase observed with IFT88 depletion. Again, no cumulative effect on the proportion of unclustered centrosomes was observed when combining IFT88 acute depletion with either dynein or HSET inhibition (Fig 2E and F). Overall, these results suggest that, as described above for IFT52, IFT88 is likely to act together with dynein and HSET to mediate efficient centrosome clustering. Of note, the effect observed on centrosome clustering using DLD-1 cells and auxin-inducible degradation of IFT88 cannot be due to impaired cilia-dependent signaling following IFT perturbation because DLD-1 cells do not grow cilia [41]. Moreover, cells are arrested in mitosis for 2 h and acutely depleted of IFT88 in mitosis, using the AID system. In this experiment, IFT88 depletion only occurs in mitosis, where cilia are absent, therefore precluding any cilia-dependent perturbation.

## IFT88/70/52/46 subcomplex directly interacts with HSET, and depletion of IFT88 reduces HSET turnover on mitotic spindle microtubules

IFT proteins were previously reported to function with dynein 1 in mitosis [33]. To further understand whether and how IFT proteins could function with the kinesin HSET during centrosome clustering, we assessed whether a subcomplex of IFT proteins including IFT52 and IFT88 (Fig 3A) could interact biochemically with HSET. We first used endogenous co-immunoprecipitation assays from MDA-MB-231 cell lysate, a cancer cell line naturally harboring supernumerary centrosomes [4], and found that IFT88 co-immunoprecipitated with HSET. This result indicates an endogenous interaction between the two proteins in cell lysates (Fig 3B). To further validate this interaction, we next expressed and purified, from insect cells, recombinant full-length (FL) GFP-HSET protein as well as the tail (Ta) and motor (Mot) domains of HSET fused to GFP (Fig 3C and D) and performed GFP-TRAP pull-down assays. Using FL GFP-HSET, we pulled down both IFT88 and IFT52 from mitotic MDA-MB-231 cell lysate (Fig 3E) confirming the interaction between the motor and IFT proteins in the cell lysate. Then, using a purified recombinant IFT-B subcomplex made of IFT88/70/52/46 (Fig 3A) we found that this interaction is direct as FL GFP-HSET can pull-down the IFT-B subcomplex *in vitro* (Fig 3F). To further identify the interaction domain of this IFT-B subcomplex on the motor, we then used either FL or truncated GFP-HSET to pull-down recombinant IFT proteins. Both FL and motor GFP-HSET interacted with IFT88 but not the tail domain (Fig 3G). This shows that the HSET/IFT protein interaction site is within the motor domain of HSET. We finally confirmed this interaction, using the motor domain truncation of HSET (aa 145–673; Fig 3C) to pull-down endogenous IFT proteins from MDA-MB-231 cell lysate (Fig 3H). In this context, HSET motor domain truncation pulled down IFT52 and IFT88 further validating the interaction. However, it did not pull-down IFT27, suggesting either that there is no interaction between IFT27 and HSET or that the amount of IFT27 pulled down is below detection level. The lack of interaction is consistent with the absence of effects of IFT27 depletion on multipolar anaphases observed in Fig 1E. Moreover, HSET motor domain truncation did not interact with IFT-A protein IFT140. This suggests that only a subset of the IFT machinery, including IFT52 and IFT88, interacts with HSET to promote centrosome clustering.

To assess whether the HSET/IFT protein interaction could affect the motor behavior, we then performed fluorescence recovery after photobleaching (FRAP) experiments. To monitor GFP-HSET turnover on mitotic spindle microtubules upon IFT88 depletion, GFP-HSET was introduced in an isogenic manner in the DLD-1 IFT88-AID cell line, described in Fig 1. As expected, GFP-HSET localized to spindle microtubules (Fig 3I). Importantly, depletion of IFT88 upon auxin addition or HSET inhibition did not affect HSET levels on the mitotic spindle before photobleaching (Fig EV3A). We thus performed FRAP experiments of GFP-HSET and analyzed its recovery along the mitotic spindle (Fig 3I and J). Upon IFT88 depletion (auxin-treated cells), we observed a 15% decrease in the maximum recovery of GFP-HSET on the mitotic spindle compared to control-treated cells (46.6% versus 55.7%, respectively). This effect, which was comparable to the decrease in recovery observed upon HSET inhibition with CW069, indicates that IFT88 depletion reduces GFP-HSET turnover on mitotic spindle microtubules.

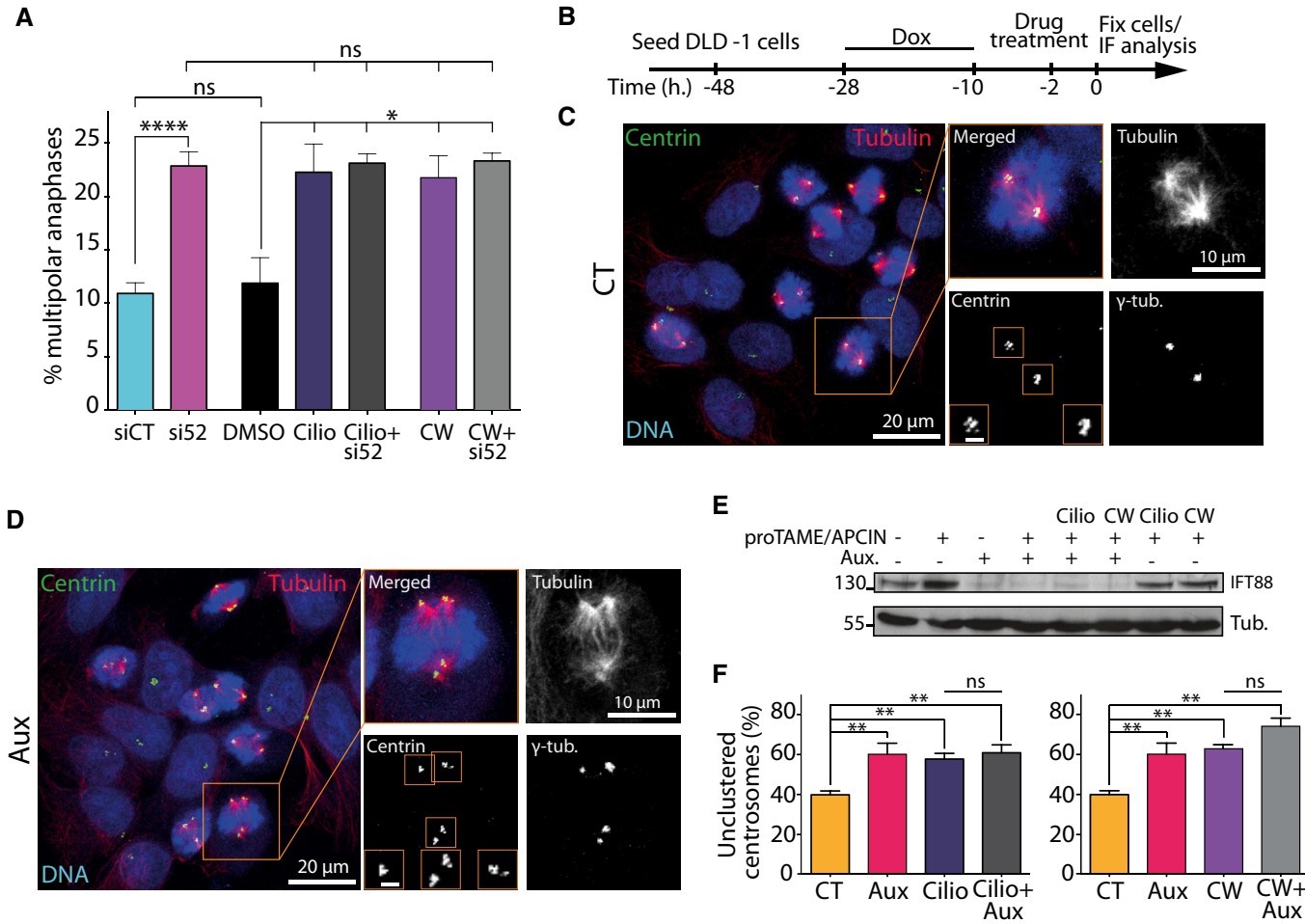

**Figure 2. IFT proteins contribute to centrosome clustering in association with the mitotic motors HSET and dynein.**

A  Quantification of the percentage of multipolar anaphases following a combination of siRNA treatment and small-molecule inhibition of dynein (ciliobrevin D: Cilio) or HSET (CW069: CW). Mean ± SEM from a minimum of three independent experiments. ****$P < 0.0001$; *$P < 0.05$; ns: non-significant; unpaired $t$-test. For each condition, a minimum of 600 cells were quantified.

B  Schematic of experimental timeline for DLD-1 cells used in panels (C–F). Cells were arrested at metaphase-to-anaphase transition using proTAME and APCIN APC/C inhibitors. Cells were also treated with or without auxin in combination with small-molecule inhibitors.

C  Representative immunofluorescence image of DLD-1 cells with supernumerary centrosomes in control condition. Scale bar in magnified box, 2 μm.

D  Representative immunofluorescence image of DLD-1 cells with supernumerary centrosomes and depleted of endogenous IFT88-AID-YFP upon auxin addition. Scale bar in magnified box, 2 μm.

E  Immunoblots showing endogenous IFT88-AID-YFP depletion upon auxin treatment combined or not with small-molecule inhibitors.

F  Quantification of the percentage of unclustered centrosomes following combination of auxin treatment and small-molecule inhibition of dynein (Cilio, left panel) or HSET (CW, right panel). Mean ± SEM from a minimum of three independent experiments. **$P < 0.01$; ns: non-significant; unpaired $t$-test. For each condition, a minimum of 233 cells were quantified.

Source data are available online for this figure.

## IFT52 depletion reduces centrosome clustering dynamics in late G2 and mitosis

Given that IFT proteins depletion results in a reduced clustering of supernumerary centrosomes and a reduced HSET motor recovery on microtubules following photobleaching, we next assessed the impact of IFT proteins depletion on the dynamics of centrosomes. We took advantage of the fact that the doxycycline-inducible Plk4 RPE-1 cells (used in Fig 1) also carried a centrin-GFP marker [34], and we imaged these cells with a high temporal resolution. In this experiment, the centrin signal corresponds to centrosomes made of a pair of centrioles (Fig EV3B). Using Imaris software, we

established the 3-dimensional trajectories of centrosomes over time and extracted parameters on their dynamics. These parameters included individual centrosome speed and inter-centrosomal distance. We analyzed centrosome trajectories from the point where all supernumerary centrosomes behaved as independent entities (last centrosome separation, blue arrows, Fig 4A, Movies EV5 and EV6) up to anaphase onset. This observation time was divided into two periods: the period from the last centrosome separation to nuclear envelope breakdown (NEB), which corresponds to late G2/prophase, and the period from NEB to anaphase onset. Measures of the duration of those two periods showed that IFT52 depletion increased the average duration of the first period,

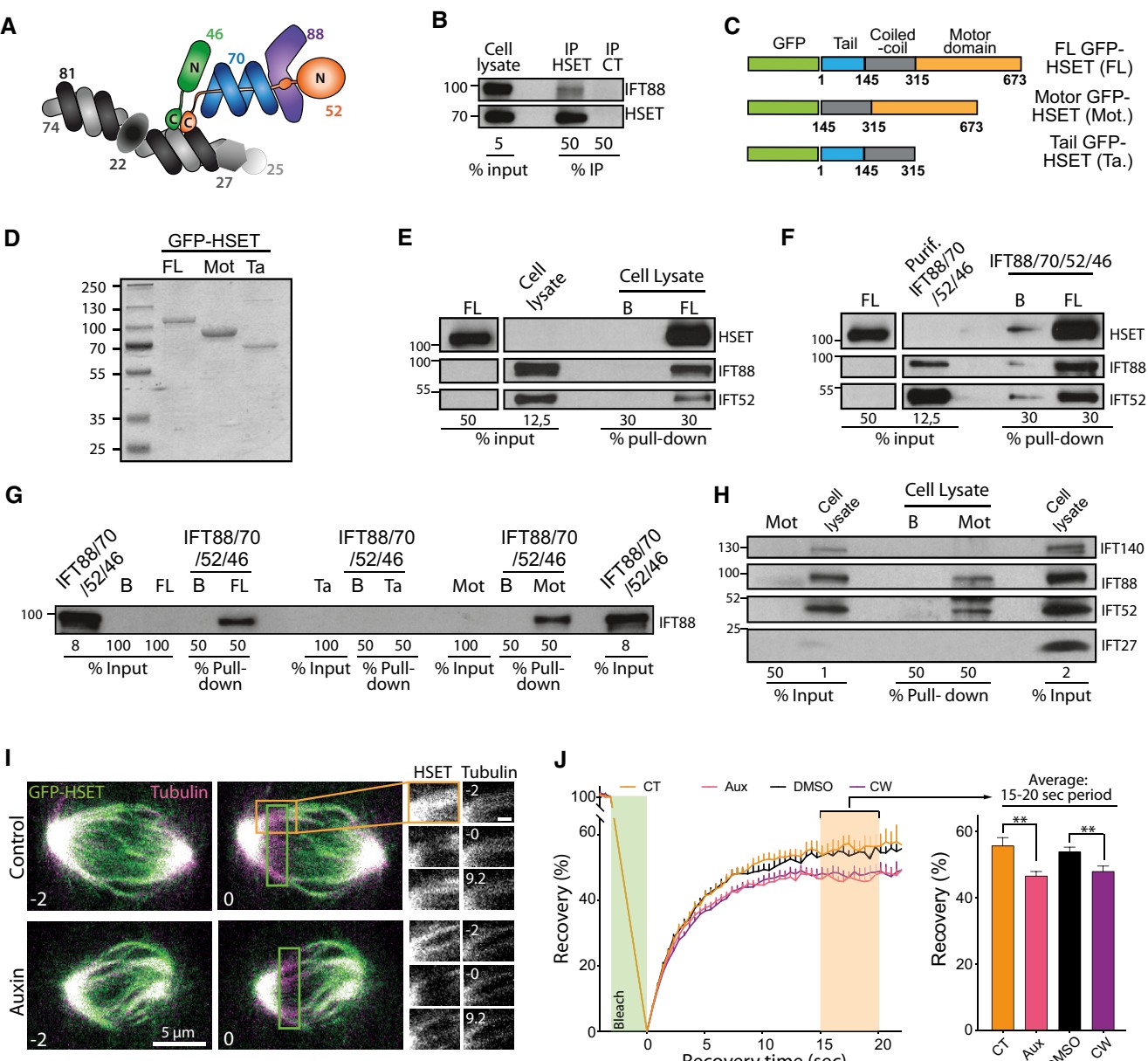

**Figure 3. IFT 88/70/52/46 complex directly interacts with HSET, and depletion of IFT88 reduces HSET turnover on mitotic spindle microtubules.**

A   Schematic representing the core of IFT-B subcomplex. Adapted from [Ref. 26], with the permission of Cold Spring Harbor Laboratory Press, © 2016. IFT proteins depicted in colors are the one for which an interaction with HSET was tested and confirmed in the following experiments (panels B–H).

B   Immunoblots of endogenous immunoprecipitation of HSET from mitotic MDA-MB-231 cell lysate.

C   Schematic of various forms of recombinant full length (FL), motor domain (Mot), and tail domain (Ta) of GFP-HSET used in panels (D–H).

D   Coomassie blue staining of the purified recombinant GFP-HSET proteins bound to GFP-trap beads as used in panels (E–H).

E   Immunoblots of a pull-down done with FL GFP-HSET and endogenous IFT proteins from a mitotic cell lysate of MDA-MB-231 cells. B: GFP-Trap beads alone. FL: GFP-Trap beads loaded with FL GFP-HSET.

F   Immunoblots of a pull-down done with FL GFP-HSET and a purified recombinant IFT complex made of IFT88, IFT70, IFT52, and IFT46.

G   Immunoblots of pull-downs done with FL, Ta, and Mot recombinant GFP-HSET, and recombinant IFT complex made of IFT88, IFT70, IFT52, and IFT46. B: GFP-Trap beads alone. FL: GFP-Trap beads loaded with FL GFP-HSET. Mot: GFP-Trap beads loaded with motor GFP-HSET. Ta: GFP-Trap beads loaded with tail GFP-HSET.

H   Immunoblots of pull-downs done with Mot GFP-HSET from a mitotic cell lysate of MDA-MB-231 cells. B: GFP-Trap beads alone. Mot: GFP-Trap beads loaded with motor GFP-HSET.

I   Representative still image of a FRAP experiment done on DLD-1 cells with endogenous IFT88 tagged with AID expressing GFP-HSET and treated with or without auxin. The green box corresponds to the photobleached area. Scale bar in magnified box, 1 μm.

J   Left: Quantification of the fluorescence recovery after photobleaching of GFP-HSET in DLD-1 cells following the indicated treatments. Mean ± SEM of four independent experiments. The green box represents the photobleaching period during the experiment. Right: Average recovery on the 15- to 20-s time period highlighted in orange on the curves (left). Mean ± SEM. **$P < 0.01$; unpaired $t$-test. For each condition, a minimum of 26 FRAP experiments were done.

Source data are available online for this figure.

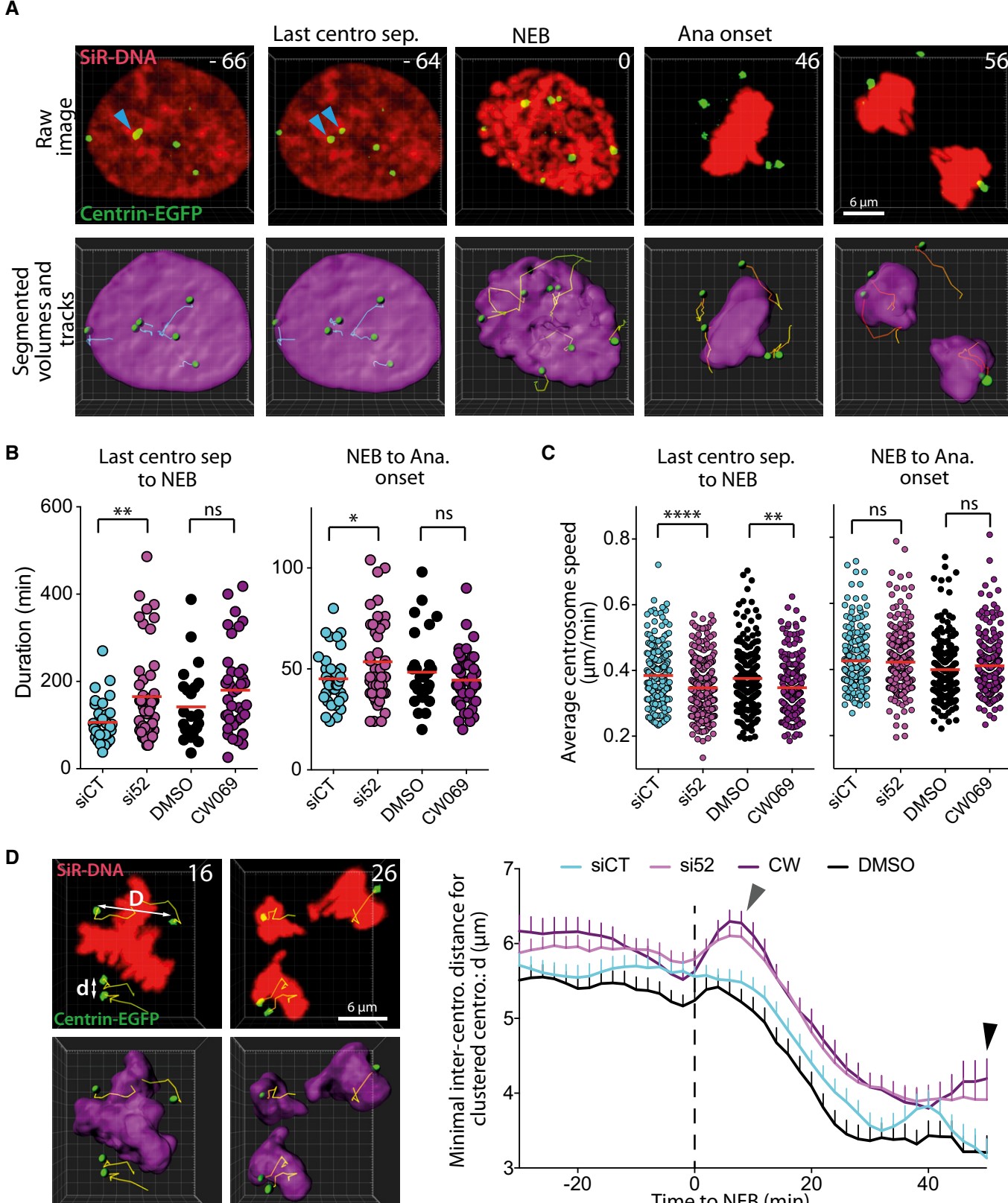

**Figure 4.**

**Figure 4.  IFT52 depletion reduces centrosome clustering dynamics in late G2 and in mitosis.**

A   Representative still images of a movie depicting centrosome movement around the nucleus in late G2 and around the condensed chromosomes in mitosis. Top row: raw microscopic images. Bottom row: segmented volumes, spots, and spot tracks generated in Imaris (Bitplane). The blue arrows indicate the last two centrosomes of the cells that separate from each other and that start to move independently. NEB: nuclear envelope breakdown. Ana: anaphase. Numbers indicate the time in minutes.

B   Quantification of the duration from last centrosome separation to NEB period (left diagram) and from NEB to anaphase onset period (right diagram) in individual cells following the indicated treatments. Red bars: mean of a minimum of three independent experiments. ns: non-significant; $**P < 0.01$; $*P < 0.05$; ns: non-significant; unpaired $t$-tests. For each condition, a minimum of 24 cells were analyzed.

C   Quantification of the average individual centrosome speed from the last centrosome separation to NEB period (left diagram) and NEB to anaphase onset period (right diagram) following the indicated treatments. Red bars: mean of three independent experiments. $****P < 0.0001$; $**P < 0.05$; ns: non-significant; unpaired $t$-tests. For each condition, a minimum of 192 centrosome tracks were analyzed.

D   Left panel, still images of a movie representing an RPE-1 cell with supernumerary centrosomes going through anaphase. In this experiment, we quantify the minimal distance between centrosomes that cluster together and move with a DNA mass at anaphase onset (d). We do not quantify the distance between centrosomes that move with different DNA mass (D). Numbers indicate the time in minutes. Right panel, quantification over time of the minimal distance between two adjacent centrosomes, in centrosomes that are clustered together at anaphase onset (d), following the indicated treatments. Mean ± SEM from a minimum of three independent experiments. The mean value for a specific time point is the centered moving average of three consecutive data, centered on the specific time point. For each condition, a minimum of 141 inter-centrosome distance tracks were analyzed. Gray arrowhead indicates a transient increase in the minimal centrosome-to-centrosome distance in CW and si52 conditions following NEB. Black arrowhead indicates the increased minimal centrosome-to-centrosome distance upon CW and si52 treatments compared to DMSO and siCT conditions at the average time of anaphase onset. Dashed line indicates the time of NEB.

compared to control conditions (106 min in siCT versus 165 in si52; Fig 4B). Inhibition of HSET using CW069 had a less pronounced effect on the duration of this first period and was not statistically significant (Fig 4B). Similarly, during the NEB to anaphase onset period the results showed that only IFT52 depletion induced a delay (45 min in siCT versus 53.5 min in si52 condition). In parallel, we measured the average velocity of individual centrosome during these two periods (Fig 4C). During the last centrosome separation to NEB period, we found that both IFT52 depletion (0.347 μm/min versus 0.384 μm/min in siCT) and HSET inhibition (0.347 μm/min versus 0.375 μm/min in DMSO) reduced centrosomes velocity compared to controls. This was not the case in the NEB to anaphase onset period upon IFT52 depletion or HSET inhibition. This effect on centrosome velocity was not due to a modification of microtubule dynamics as IFT52 depletion did not affect microtubule dynamics monitored by tracking EB1 comet velocity (Fig EV3C). To dynamically quantify the impact of IFT52 and HSET on centrosome clustering, we next studied if IFT52 depletion or HSET inhibition resulted in an increased distance between clustered centrosomes. We measured the minimal distance between adjacent centrosomes that would eventually be clustered together at anaphase onset: "d" in Fig 4D, as opposed to distance "D" which is the minimal distance between two centrosomes that end up unclustered at anaphase onset (Fig 4D and Movie EV7). We found that both IFT52 depletion and HSET inhibition resulted in an increased distance between the closest centrosomes during the observation period: last 30 min before NEB up to 50 min after NEB (50 min is the average time of anaphase onset for the four conditions). More specifically, at 50 min (black arrowhead Fig 4D), we observed a 20% increase in this distance for si52 versus siCT conditions (3.9 μm versus 3.1 μm) and a 24% increase between CW069 and DMSO conditions (4.2 μm versus 3.2 μm). This increase in inter-centrosomal distance correlates with the increase in multipolar anaphases observed in the same cells with similar treatments (Fig 1). Collectively, the data on centrosome dynamics indicate that IFT52 depletion results in an increased distance between centrosomes in mitosis that correlates with an increased proportion of multipolar mitotic figures in anaphase.

## Cancer cells with natural centrosome amplification rely on IFT88 for efficient clustering and proliferation

Having demonstrated that a subset of IFT proteins contributes to the clustering of supernumerary centrosomes generated by the overexpression of Plk4 kinase or by cell tetraploidization (Figs 1 and EV1 and EV2), we assessed if IFT proteins were also necessary for centrosome clustering in cells naturally harboring supernumerary centrosomes. Cancer cells frequently harbor supernumerary centrosomes, so we screened the level of centrosome amplification in four different cancer cell lines: 786-0, renal adenocarcinoma cells (18.3% centrosome amplification); HT-29, colorectal adenocarcinoma cells (19.3% amplification); MDA-MB-231, breast adenocarcinoma cells (27.1% amplification); and Caco-2, colorectal adenocarcinoma cells (36.6% amplification; Figs 5A and B, and EV3E). Then, we quantified the proportion of unclustered centrosomes in mitosis, in those cells, following IFT88 depletion (Fig 5C and D). We found that in all four cell lines, depletion of IFT88 led to an increased proportion of unclustered centrosomes (786-O, 40.8% siCT versus 58.8% si88; HT-29, 43.7% versus 69.7%; MDA-MB-231, 43.3% versus 66.2%; and Caco-2, 52.5% versus 81.1%). This result indicates that in cells naturally harboring supernumerary centrosomes, IFT88 is also required for efficient centrosome clustering. Importantly, depletion of IFT52 also results in an increased proportion of unclustered centrosomes in Caco-2 cells, a phenotype that was rescued by re-expressing mouse IFT52 (Figs 5E and EV3D). To control that the perturbation of centrosome clustering resulted in multipolar anaphases, we took advantage of a MDA-MB-231 cell line expressing H2B-EGFP and analyzed live mitosis in binucleated cells (Fig 5F–H). These binucleated cells, likely generated after cytokinesis failure, are naturally present in the cell population. Of note, these binucleated cells naturally harbor a higher percentage of supernumerary centrosomes compared to the rest of the population, which is mononucleated. Indeed, 67.1% of those binucleated cells have supernumerary centrosomes compared to 27.1% when considering all cell types in the MDA-MB-231 cell population (Fig 5A). Upon IFT88 depletion the proportion of multipolar anaphases in binucleated cells went from 12.1 to 37.2%, clearly indicating that IFT88 depletion was triggering multipolar anaphases. Multipolar

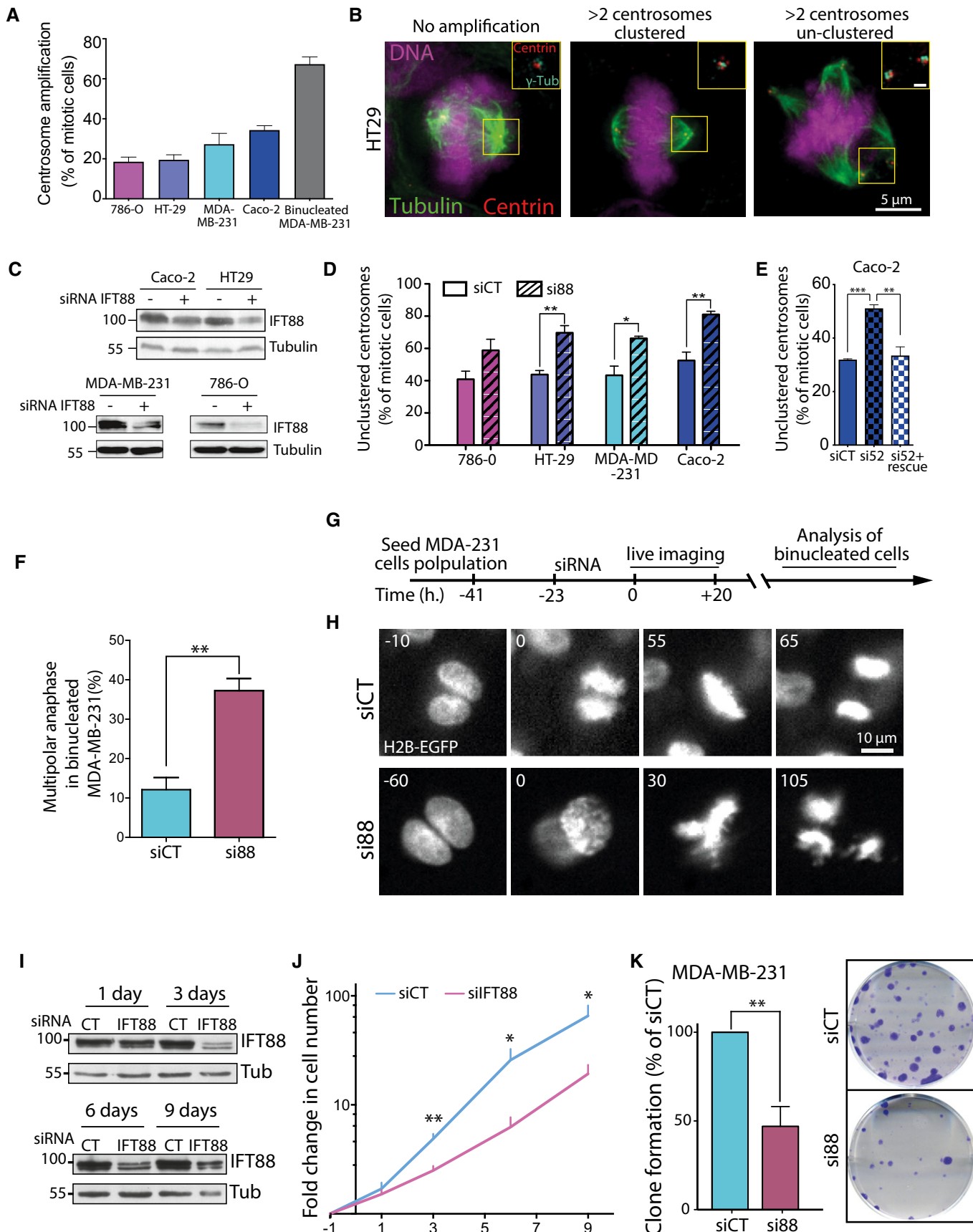

**Figure 5.**

**Figure 5.  Cancer cells naturally harboring supernumerary centrosomes depend on IFT proteins for efficient centrosome clustering and proper proliferation.**

A   Quantification of "natural" centrosome amplification in various mitotic cancer cell lines. For 786-O, HT-29, MDA-MB-231, and Caco-2 cells, the mean ± SEM from 4 independent experiments is represented and a minimum of 485 cells were analyzed. For MDA-MB-231 binucleated cells, the means ± SEM from four independent experiments are represented, and 100 binucleated cells within the general MDA-MB-231 cell population were analyzed.

B   Immunofluorescence images of mitotic figures of HT-29 colorectal cancer cells with normal and abnormal centrosome numbers. Scale bar in magnified box, 1 μm.

C   Immunoblots of various colorectal cell lysates upon IFT88 depletion.

D   Quantification of unclustered centrosomes figures in cells with centrosome amplification upon control or IFT88 depletion. MDA-MB-231 and Caco-2 cells were treated with MG132 for 2 h in order to enrich for clustered centrosomes both in control and si88 conditions. Mean of three independent experiments ± SEM. **$P < 0.01$; *$P < 0.05$; unpaired t-test. For each condition, a minimum of 150 cells were quantified.

E   Quantification of cells with unclustered centrosomes in Caco-2 cells with centrosome amplification upon control or IFT52 depletion. Cells were treated with MG132 for 2 h in order to enrich for clustered centrosomes both in control and si52 conditions. Mean of three independent experiments ± SEM. ***$P < 0.001$; **$P < 0.01$; unpaired t-test. For each condition, a minimum of 150 cells were quantified.

F   Quantification from live analysis of multipolar anaphases in binucleated MDA-MD-231 cells. Mean of a minimum of three independent experiments ± SEM. **$P < 0.01$; unpaired t-tests. A minimum of 60 mitosis of binucleated cells were analyzed for each condition.

G   Schematic of experimental timeline on MDA-MB-231 used for binucleated analysis. Cells were treated with siCT or si88 overnight. The quantification of multipolar anaphases was done only on binucleated cells.

H   Representative images from a time lapse of binucleated MDA-MB-231 cells treated or not with a siRNA against IFT88.

I    Immunoblots of MDA-MB-231 cell lysate up to 9 days after siRNA depletion. Tub: tubulin; loading control.

J    Quantification of the change in cell number in MDA-MB-231 cells along 9 days in si88 and siCT conditions. Mean of six independent experiments ± SEM. **$P < 0.01$; *$P < 0.05$; unpaired t-tests.

K   Left panel, quantification of the clonogenic capabilities of MDA-MB-231 cells in siCT and si88 conditions. siCT is normalized to 100%. Mean of four independent experiments done in duplicates ± SEM. **$P < 0.01$; unpaired t-tests. Right panel, representative images of the clonogenic assays stained with crystal violet.

Source data are available online for this figure.

anaphases are known to result in severe aneuploidy and cell death in the following cell cycles [4]. Therefore, we then tested how IFT88 depletion was affecting MDA-MB-231 cell proliferation over the course of 9 days (Fig 5I and J). We found that IFT88 depletion significantly reduces cell proliferation starting from day three and up to day nine. In an alternative way to test cell growth perturbation, we performed clonogenic assays with the MDA-MB-231 cells and we found that depletion of IFT88 results in a twofold decrease in the ability of the cells to form clones (Fig 5K). Similar defects on clonogenic properties were also observed in cancer cells, in which supernumerary centrosomes were induced either by tetraploidization (HCT-116) or Plk4 overexpression (DLD-1), following knockdown of IFT88 (Figs EV1A and EV2B–E and EV3F–I).

Overall, we show here that a subset of IFT proteins contributes to centrosome clustering through their interaction with the mitotic motors dynein, previously identified to interact with IFT proteins [33], and with HSET (Figs 1–3 and EV4A). Detailed analysis of centrosome dynamics following IFT proteins perturbation also revealed that IFT proteins impact these dynamics starting from late G2/prophase, in part through their interaction with the kinesin HSET (Figs 4 and EV4B). At anaphase onset, IFT52 contributes to the maintenance of a close distance between supernumerary centrosomes that have to be clustered at the pole. Thus, at this stage, the absence of IFT52 weakens centrosome clustering, a defect that eventually results in an increase in multipolar anaphases (Figs 1, 2, 4 and EV4C).

Altogether, these data introduce IFT proteins as new players in the regulation of centrosome dynamics in mitosis by showing that IFT52 is required to maintain a close distance between supernumerary centrosomes. We observed a delay in mitotic progression (last centrosome separation to NEB and NEB to anaphase onset) in IFT52-depleted condition but not with CW069 perturbation. This delay could result from a delay in the assembly of the mitotic spindle. Indeed, IFT88 depletion was recently shown to be necessary for the timely insertion of the newly generated k-fiber minus ends into the mitotic spindle and for proper chromosome alignment [31].

Therefore, because IFT52 is essential for the stability of the IFT-B subcomplex that includes IFT88 [25,42], it is possible that its depletion also triggers delays in spindle assembly and chromosome alignment which eventually results in an increased mitotic progression duration. This delay in spindle assembly, which is a dynamic event, does not impact the global spindle morphology observed in immunofluorescence images of the RPE-1 cells (Fig EV1). The observation that the simultaneous IFT52 depletion and HSET inhibition do not affect centrosome speed observed during NEB to anaphase onset period indicates that HSET is not, at this stage, the dominant force in generating centrosome speed. However, the increase in inter-centrosomal distance, observed with the same perturbations, during the same mitotic period, indicates that HSET is still important for centrosome clustering. Interestingly, the range of distance measured between centrosomes that are clustered at anaphase onset (from 5.6 μm at NEB to 3.1 μm at anaphase onset) is within the range of distance where the kinesin HSET is acting through its motor activity to promote centrosome clustering, as recently demonstrated in MCF10a cells [20]. Finally, monitoring of centrosome dynamics also revealed a sharp and transient increase in the inter-centrosomal distance following NEB in the perturbed conditions (gray arrowhead Fig 4D). This peak in the inter-centrosomal distance upon NEB could result from the loss of centrosome attachment to the nuclear envelope mediated by dynein [43] or CENP-F [44] which is not counteracted by HSET activity when it is perturbed in si52 and CW069 conditions. This analysis therefore shows that dynein and HSET have complementary activities in the positioning of centrosome at mitosis onset and that HSET, together with IFT proteins, is essential to maintain centrosome cohesion at NEB when the link between centrosomes and the nuclear envelope mediated by dynein is lost (Fig EV4B). Overall, this work brings novel insights on the dynamics of centrosome movements during mitosis.

Centrosome clustering and centrosome dynamics are highly dynamic events that require a robust machinery to function properly [13,20]. Mechanistically, we show that IFT proteins are required for

both processes in part through their interaction with HSET. Of note, IFT88 depletion recapitulates the effect of HSET inhibition, with CW069, regarding its recovery on microtubules (Fig 3I and J). CW069 inhibits the ATPase activity of HSET [39], and it is known that ATP hydrolysis is required for the conformational change in the neck domain responsible for the power stroke of kinesin-14 motors such as HSET. This conformational change induces the subsequent release of the motor head from the microtubule [45–47] resulting in a reduced turnover of HSET on the microtubule. The similarity between the chemical inhibition and IFT88 depletion regarding HSET recovery in the FRAP experiment suggests that IFT88 binding to HSET could impact the bio-mechanical capabilities of HSET. Kinesins can be autoinhibited due to the interaction between their motor domain and tail domain [48]. This inhibition can be released through the binding of a partner protein as shown for kinesin 1 motor [49]. Conversely, kinesin 14 family member KCPB, a plant kinesin related to HSET, is known to be inhibited by the regulating partner protein KIC [50]. Considering those regulatory mechanisms of kinesins by partner proteins, it is possible that IFT proteins can activate HSET by competing with inhibiting regulators of HSET motor or through the release of an autoinhibition. This latter mechanism is strongly supported by a recent study showing the activation of kinesin 2, *in vitro,* by an IFT-B subcomplex proteins containing IFT46-IFT52-IFT70-IFT88 [37]. Elucidating the role of IFT proteins in regulating the activity of HSET will be the aim of future works. This will include a thorough *in vitro* biochemical study in order to reveal whether and how each individual IFT protein and/or subcomplexes of IFT proteins interact with specific HSET domains and to understand how these interactions affect HSET motor binding and motor activities on microtubules.

Finally, we show that cancer cells naturally harboring supernumerary centrosomes are also dependent on IFT proteins for their sustained proliferation (Fig 5). Therefore, elucidating if and potentially how IFT proteins directly regulate the kinesin HSET or other mitotic motor activities will be of particular interest for future studies. Indeed, the characterization of a novel regulatory interaction of HSET may provide the required knowledge for the development of novel therapeutic strategies, as perturbation of this interaction should selectively target and kill cancer cells with supernumerary centrosomes.

# Materials and Methods

### Cell lines and cell culture conditions

Cells were maintained at 37°C in a 5% $CO_2$ atmosphere. Human telomerase-immortalized retinal pigment epithelial (RPE-1) cells were cultured in DMEM/F-12 (1:1) medium supplemented with 10% FBS, 100 U/ml penicillin–streptomycin, and 2 mM L-glutamine. Generation of the RPE-1 cells carrying centrin-GFP and the tetracycline-inducible system for Plk4 is described in [Ref. 34]. Stable expression of H2B-EGFP-IRES-mcherry-tubulin was achieved following retroviral transduction and isolation of a single clone using fluorescence-activated cell sorting. RPE-1 cells stably expressing EB1-GFP were a gift from S. Doxsey (UMASS Medical School). HCT-116, DLD-1, MDA-MB-231, and HT29 were cultured in DMEM with 10% FBS, 100 U/ml penicillin–streptomycin, and 2 mM L-glutamine.

Tetraploid HCT-116 cells were generated as described in [Ref. 51]. Caco-2 cells were cultured in DMEM with 20% FBS, 100 U/ml penicillin–streptomycin, and 2 mM L-glutamine. 786-O cells were cultured in RPMI 1640 with 10% FBS, 100 U/ml penicillin–streptomycin, and 2 mM L-glutamine. HT-29, Caco-2, and 786-O cells were obtained from IRCM Cell Culture Unit, Montpellier.

### Generation of DLD-1 IFT88-AID-YFP and derived cell lines

DLD-1 IFT88-AID-YFP cells were generated by adding an auxin-inducible degron (AID) tag followed by a YFP tag at the 3′ end of the last exon on the IFT88 genomic locus. In detail, DLD-1 Flip-In T-Rex cells stably expressing TIR1-9xMyc protein [36] were used for the targeting. sgRNA targeting two regions adjacent to the 3′ end of IFT88 gene were introduced under the control of U6 transcription promoter into two separated vectors encoding for the expression of the Cas9 nickase (D10A) [52] (Addgene 42335). A donor construct containing ≈ 600 bp recombination arms surrounding the 3′ end of IFT88 locus, in frame with a sequence encoding for an AID-YFP-Stop sequence, was generated. All three vectors were transfected into the DLD-1 Flip-In T-Rex TIR1 cells using X-tremeGENE 9 DNA Transfection Reagent (Roche). Cells were sorted based on their YFP fluorescence, and single clones were isolated. Homozygous targeted clones were identified by PCR. Targeting of IFT88 and degradation of IFT88-AID-YFP were confirmed by Western blot following addition of auxin (Sigma-Aldrich) at 500 μM in the culture medium for 1 h.

The generation of DLD-1 IFT88-AID-YFP cells expressing stable inducible hPlk4 was done using hPlk4-YFP cloned into pCDNA5-frt-TO (gift from A.J. Holland) introduced into the cells via FRT/Flp-mediated recombination [53]. Cells were transduced with a retroviral construct expressing H2B-EGFP-IRES-mcherry-tubulin, and single clones were isolated using fluorescence-activated cell sorting.

To generate DLD-1 IFT88-AID-YFP cells expressing stable GFP-HSET, human HSET cDNA (Ultimate ORF clone # IOH4703, Life Technologies) was cloned into pCDNA5-3xFlag-GFP using the Gateway system. GFP-HSET was introduced into the cells via FRT/Flp-mediated recombination [53] in a unique FRT site and, therefore, is expressed in an isogenic manner in the cell population.

### siRNA and cDNA transfections

Targeted proteins were depleted with small-interfering RNAs (siRNAs) designed and ordered from Dharmacon. siRNAs sequences include: non-targeting (siCT): UGGUUUACAUGUCGACUAA (D-0018110-01); human IFT27: CAGAAAAGCUACACCCUGA (J-009565-07); human IFT52: UAUCAAAGCGGAAUCGAGA (J-020994-20); and human IFT88: CGACUAAGUGCCAGACUCAUU [33]. SiRNAs were delivered to cells at a final concentration of 100 nM using Lipofectamine RNAiMAX (Invitrogen) according to manufacturers' instructions. For rescue experiments, mouse IFT52 cDNA transfection was done with 250 ng pCDNA3-mcherry-IFT52 cDNA using X-tremeGENE 9 (Roche) according to manufacturers' instructions.

### Antibodies and chemicals

The following primary antibodies were used (Western blot, WB; immunofluorescence, IF). Proteintech: IFT88 (13967-1-AP; WB: 1/500; IF: 1/250), IFT52 (17534-1-AP; WB: 1/500; IF: 1/500), IFT81

(11744-1-AP; WB: 1/250), IFT140 (14460-1-AP; WB: 1/500); Novus Biological: IFT27 (NBP1-87170, WB: 1/200); Sigma-Aldrich: α-tubulin (T6169, WB: 1/1,000; IF: 1:1,000), FITC-conjugated α-tubulin (F2168, IF: 1/300), GAPDH (G8795, WB: 1/10,000); Abcam: α-GFP-FITC (ab6662, IF: 1/500), and HSET/KIFC1 (ab55388, WB: 1/1,000); Cell Signaling Technology: KIFC1/HSET (12313, IP: 1/100); Millipore: centrin (20H5, IF: 1/5,000); Home-made: γ-tubulin (HM2569, gift from S. Doxsey laboratory [54], IF: 1/500); Homemade: polyglutamylated tubulin (GT-335, gift from C. Janke Lab [55]) and DAPI (Cell Signaling, IF 1/10,000). Secondary antibodies include for IF: Alexa Fluor 488-, 568-, or 647-conjugated anti-rabbit or anti-mouse secondary antibodies (Molecular Probes, 1/1,000), and for WB: anti-mouse and anti-rabbit IgG, HRP-linked antibody (Cell Signaling, 1/5,000 and 1/10,000, respectively).

The following chemicals were used in the different experiments: p38 inhibitor at 10 μM (Selleckchem SB302580) to allow for continuous proliferation of RPE-1 cells following centrosome amplification [20]; nocodazole, 100 ng/ml (Sigma-Aldrich); MG132, 10 μM (Sigma-Aldrich) HSET inhibitor CW069 at 150 μM with DLD-1 and 25 μM with RPE-1 cells (Selleckchem, S7336); dynein inhibitor ciliobrevin D, at 50 μM with DLD-1 and 20 μM with RPE-1 cells (Calbiochem, 250401); APC/C inhibitors ProTAME 10 μM (Boston-Biochem, I-440) and APCIN 40 μM (BostonBiochem, I-444); doxycycline, 1 μg/ml (Sigma-Aldrich, D9891); and auxin, 500 μM (Sigma-Aldrich, I5148).

## Gel electrophoresis and immunoblotting

Samples were separated by SDS–PAGE, transferred onto PVDF membrane using Mini Trans-Blot cells (Bio-Rad), and probed overnight at 4°C using the appropriate antibodies. Western Lightning Plus-ECL Kit (PerkinElmer) and Blue Devil (Dominique Dutscher) films were used to reveal the Western blots.

## Immunofluorescence

Cells were fixed in −20°C MeOH for immunofluorescence experiments to preserve microtubule staining. Then, cells were blocked with PBS-BSA 1%-Triton 0.5% and stained for immunofluorescence with the appropriate primary and secondary antibodies. Slides were mounted in ProLong Gold (Life Technologies).

## Microscopy and image analysis

Epifluorescence images were acquired with a Leica DM6000 Microscope (objective: 63× Plan Apo 1.4 NA) equipped with a CoolSNAP HQ2 Camera and controlled by MetaMorph (Molecular Devices). Confocal images were acquired with a Zeiss LSM780 microscope (objective: 63× Plan Apo 1.4 NA) controlled by ZEN software (Zeiss); with a spinning disk confocal microscope, a Nikon Ti Eclipse coupled to a Yokogawa CSU-X1 spinning disk head and an EMCCD iXon Ultra camera (objectives: 60× Plan Apo 1.4 NA and 100× Plan Apo 1.45 NA), controlled by the Andor iQ3 software (Andor); or with a Dragonfly spinning disk microscope (Andor) equipped with iXon 888 Life EMCCD Andor camera (100× Plan Apo 1.45 NA) and controlled by Fusion software (Andor, Oxford instruments).

Live-cell imaging for quantification of multipolar anaphases was performed, overnight at 37°C in a $CO_2$-controlled atmosphere, using

an inverted Olympus IX83 microscope equipped with a Zyla 4.2 MP sCMOS camera (objective: 40× LUCPLFLN 0.6NA RC2, air) and controlled by MetaMorph (Molecular Devices) on cells seeded in 12-well plates. Images were acquired every 5 min on three plans with 8-μm Z interval. DNA was visualized with the H2B-EGFP signal or with SiR-DNA probe (Spirochrome) at 500 nM added 1 h before imaging begins. Image processing and analysis (cropping, rotating, brightness, contrast adjustment, color combining, and measurements) were performed using Fiji (ImageJ). Live imaging of centrosome clustering dynamics was performed, overnight at 37°C in a $CO_2$-controlled atmosphere, using spinning disk confocal microscope, a Nikon Ti Eclipse coupled to a Yokogawa CSU X1 spinning disk head and an EMCCD iXon Ultra camera (objective: 40× Plan Apo 1.3 NA) controlled by the Andor iQ3 software (Andor) on cells seeded on μ-Slide 8 Well (ibiTreat, Ibidi). Images were acquired every 2 min on 23 plans with 0.5-μm Z interval. DNA was visualized using SiR-DNA probe (Spirochrome) at 500 nM and centrosomes with centrin-GFP signal. Image segmentation and centrosome tracking were performed using Imaris (XT module, Bitplane). Individual cell drift and rotation were corrected based on the DNA mass trajectories. Spots were created for each individual centrosome using the spot-tracking tool, and various parameters on centrosome spot dynamics were generated using the XT module.

## FRAP experiments

FRAP was performed on GFP-HSET, DLD-1 IFT88-AID-YFP cell line. In the parental cell line (IFT88-AID-YFP without GFP-HSET), no signal from IFT88-AID-YFP is detectable on the GFP channel under the FRAP imaging conditions. Cells were seeded 24 h before the experiment in 96-square well ibiTreat bottom plate (Ibidi). Two hours before imaging, auxin (500 μM) was added in the corresponding condition. Thirty minutes before imaging, SiR-tubulin Cy5 at 1 μM and verapamil at 10 μM were added to the medium to visualize microtubules according to manufacturer's instructions (Spirochrome). CW069 at 150 nM (Selleckchem) or DMSO was also added 30 min before imaging on the corresponding conditions. Images were collected at 37°C with a Zeiss LSM780 confocal microscope (objective: 63×/1.4 NA DIC Plan Apo). FRAP experiments were performed using the 488 nm laser line controlled by ZEN software (Zeiss). Images were acquired on a single plan every 460 ms, and 10 measurements were carried out before the bleaching event. The bleached zone was a fixed area represented by the green square in Fig 3I. FRAP analysis was performed, using ImageJ, only in cells where microtubules remain stable in focus during the acquisition time in the bleached zone. Background fluorescence was measured from a zone outside the spindle, and observational photobleaching was measured from an area equivalent to the FRAP zone in a zone not submitted to FRAP. Photobleaching and background normalization were calculated using Excel software (Microsoft) and applied to the mean fluorescence intensity values in the FRAP zone.

## Clone formation and cell proliferation assays

Cells were treated with siRNA as indicated. The next day, for DLD-1 and HCT-116 cells, 400 cells were seeded in a 10-cm dish, and for MDA-MB-231, 100 cells were seeded in a 6-well plate. Medium was replaced every other day. After 14 days, colonies were fixed for 10

min in methanol and stained for 10 min using a crystal violet staining solution (1% crystal violet and 20% ethanol). For MDA-MB-231 cell proliferation assay, $1 \times 10^5$ cells were seeded in a 6-well plate. The next day, they were treated with siRNA as indicated. Cells were trypsinized and counted, in duplicate, and re-seeded at the indicated time points, and cell lysates for Western blots were generated in parallel, with the same setup, at the same time points.

**Recombinant protein purifications, GFP-TRAP pull-down assays, and immunoprecipitations**

Recombinant subcomplex of IFT proteins IFT88/70/52/46 was expressed in bacteria and purified as described in Ref [42]. FL and truncated GFP-HSET were expressed in sf9 insect cells and purified as described in [Ref. 56]. Briefly, full-length N-terminal hexa-histidine-tagged GFP-HSET, GFP-HSET-tail, and GFP-HSET-motor cloned in pOET1C-modified vectors (gift of S. Diez, TU Dresden) were expressed in sf9 insect cells using the flashBAC system (Oxford Expression Technologies). Cells were harvested, snap-frozen in liquid nitrogen, and stored at −80°C until purification. Cell pellets were resuspended in purification buffer (50 mM sodium phosphate buffer, pH 7.5, 1 mM $MgCl_2$, 10 mM 2-mercaptoethanol, 300 mM NaCl, 0.1% Tween-20/v, 10% glycerol w/v, 30 mM imidazole, and EDTA-free protease inhibitors (Roche)). Crude lysate was centrifuged at 20,000 g at 4°C and loaded on Ni-NTA resin (Qiagen). The resin was washed extensively (3 × 1 h) with purification buffer containing 30 mM imidazole, plus one wash with 50 mM imidazole. Proteins were eluted in purification buffer containing 200 mM imidazole, dialyzed overnight against storage buffer (50 mM Hepes, pH 7.5, 200 mM NaCl, 1 mM $MgCl_2$, DTT 1 mM, MgATP 0.5 mM), aliquoted, snap-frozen in liquid nitrogen, and stored at −80°C. For GFP pull-down assays with purified HSET proteins, purified HSET variants were resuspended in pull-down buffer (80 mM PIPES, 2 mM $MgCl_2$, 1 mM EGTA, 250 mM NaCl, 1 mM DTT, 5% FBS, 1× protease inhibitors (Roche)). 40 µg of proteins was incubated with 20 µl of GFP-Trap beads (Chromotek), for 90 min at 4°C and 30 min at RT. They were washed 2 times with 1 ml of pull-down buffer containing 0.2% NP-40. For pull-down of IFT88/70/52/46, 20 µl of GFP-Trap beads with either FL, tail, or motor GFP-HSET was incubated with 300 µl of recombinant IFT proteins for 1 h at RT. Beads were washed five times with 800 µl of buffer containing 0.2% NP-40 and resuspended in Laemmli buffer for SDS–PAGE analysis. For pull-down in cell lysate, mitotic MDA-MB-231 cells were used. Three 15-cm plates of MDA-MB-231 cells at 40% confluency were treated with nocodazole (100 ng/ml) for 18 h. On the day of the experiment, mitotic-arrested cells were collected by shake-off. Cell lysate was made by resuspending $1 \times 10^7$ cells in 500 µl of lysis buffer (50 mM Hepes, pH 7.5, 150 mM NaCl, 1.5 mM $MgCl_2$, 1 mM EGTA, 1× protease inhibitor (Roche), 0.5% NP-40) incubated for 30 min at 4°C with frequent pipetting. Lysate was spun 10 min at 21,000 g. Clear lysate was adjusted to 0.2% NP-40 with lysis buffer without NP-40. GFP-Trap with the GFP-HSET constructs were incubated with 300 µl of cell lysate for 1 h at RT, washed five times with 800 µl of buffer containing 0.2% NP-40, and resuspended in Laemmli buffer for SDS–PAGE analysis. For endogenous immunoprecipitations, MDA-MB-231 cell lysate was prepared as for GFP pull-down assay. Rabbit IgG or KIFC1/HSET antibody (Cell Signaling Technology, ref. #12313) was added to the cell lysate and incubated overnight at 4°C. Lysates were then incubated for 1 h with protein G-PLUS agarose beads (Santa Cruz; sc-2002). Beads were washed four times with 500 µl lysis buffer containing 0.2% NP-40, and the immunoprecipitated proteins were separated by SDS–PAGE and analyzed by Western blotting.

**Statistical analysis**

The number of cells counted per experiment for statistical analysis is indicated in figure legends. Graphs were created using GraphPad Prism software. P-values were calculated using two-tailed Student's t-tests.

Expanded View for this article is available online.

## Acknowledgements

The experiments were performed on the imaging platform Montpellier Ressources Imagerie (MRI), Montpellier, member of the national infrastructure France-BioImaging supported by the French National Research Agency (ANR-10-INSB-04, "Investments for the future"). We particularly thank S. De Rossi for his assistance regarding image analysis, M. Boyer for her help with cytometry, and C. Hassen-Kodja for his help with data analysis. We thank the Montpellier Genomic Collection Platform (S. Fromont and F. Lionneton). We would like to thank D. Xirodimas for critical reading of the manuscript. We also thank A. Holland, D. Fachinetti, S. Diez, S. Doxsey, and C. Janke for reagents and advice. We thank the IRCM Cell Culture Unit (Grant INCa-DGOS-Inserm 6045) for cancer cell lines. This work was supported by Projet Fondation ARC grants (PJA 20151203324 and PJA 20171206441 to BV), ANR JCJC IRMM grant (ANR-18-CE11-0025-01 to BV), ANR "Chaire d'excellence" CilMitoCyst grant (ANR-12-CHEX-005 to BD), the Marie Curie Career Integration Grant (CilMitoPatho to B.D.), the Fondation pour la Recherche Médicale (Partenariat Fondation Schlumberger pour l'Education et la Recherche to B.D.) and the CNRS (BV, CA, and BD).

## Author contributions

BV, NT, AG, ADou, ADos, MC, JM, SH, and CA conducted the experiments and analyzed the experimental work. MT and EL generated the recombinant IFT proteins. BV and BD conceived, designed, and supervised the project. BV and BD assembled the figures and wrote the manuscript.

## Conflict of interest
The authors declare that they have no conflict of interest.

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
