## [Review Process File · EMBO Reports]

IFT proteins interact with HSET to promote supernumerary centrosome clustering in mitosis

Benjamin VITRE, Nicolas TAULET, Audrey GUESDON, Audrey DOUANIER, Aurelie DOSDANE, Melanie CISNEROS, Justine MAURIN, Sabrina HETTINGER, Christelle ANGUILLE, Michael TASCHNER, Esben LORENTZEN and Benedicte DELAVAL

Review timeline:

Submission date:	6 September 2019
Editorial Decision:	11 October 2019
Revision received:	24 January 2020
Editorial Decision:	14 February 2020
Revision received:	25 February 2020
Accepted:	12 March 2020

Editor: Deniz Senyilmaz-Tiebe

Transaction Report:

1st Editorial Decision

11 October 2019

Thank you for the submission of your research manuscript to our journal, which was now seen by three referees, whose reports are copied below.

As you can see, the referees express interest in the proposed role of IFT in centrosome clustering. However, they also raise a number of concerns that need to be addressed to consider publication here.

Given these constructive comments, we would like to invite you to revise your manuscript with the understanding that the referee concerns (as detailed above and in their reports) must be fully addressed and their suggestions taken on board. Please address all referee concerns in a complete point-by-point response. Acceptance of the manuscript will depend on a positive outcome of a second round of review. It is EMBO reports policy to allow a single round of revision only and acceptance or rejection of the manuscript will therefore depend on the completeness of your responses included in the next, final version of the manuscript.

Supplementary/additional data: The Expanded View format, which will be displayed in the main HTML of the paper in a collapsible format, has replaced the Supplementary information. You can submit up to 5 images as Expanded View. Please follow the nomenclature Figure EV1, Figure EV2 etc. The figure legend for these should be included in the main manuscript document file in a section

called Expanded View Figure Legends after the main Figure Legends section. Additional Supplementary material should be supplied as a single pdf labeled Appendix. The Appendix includes a table of content on the first page with page numbers, all figures and their legends. Please follow the nomenclature Appendix Figure Sx throughout the text and also label the figures according to this nomenclature. For more details please refer to our guide to authors.

- 1) a .docx formatted version of the manuscript text (including legends for main figures, EV figures and tables). Please make sure that the changes are highlighted to be clearly visible.
- 2) individual production quality figure files as .eps, .tif, .jpg (one file per figure).
- 3) a .docx formatted letter INCLUDING the reviewers' reports and your detailed point-by-point responses to their comments. As part of the EMBO Press transparent editorial process, the point-by-point response is part of the Review Process File (RPF), which will be published alongside your paper. For more details on our Transparent Editorial Process, please visit our website: <https://www.embopress.org/page/journal/14693178/authorguide#transparentprocess>
You are able to opt out of this by letting the editorial office know (emboreports@embo.org). If you do opt out, the Review Process File link will point to the following statement: "No Review Process File is available with this article, as the authors have chosen not to make the review process public in this case."
- 4) a complete author checklist, which you can download from our author guidelines (<http://embor.embopress.org/authorguide>). Please insert information in the checklist that is also reflected in the manuscript. The completed author checklist will also be part of the RPF.
- 5) Please note that all corresponding authors are required to supply an ORCID ID for their name upon submission of a revised manuscript (<https://orcid.org/>). Please find instructions on how to link your ORCID ID to your account in our manuscript tracking system in our Author guidelines (<http://embor.embopress.org/authorguide>).
- 6) We replaced Supplementary Information with Expanded View (EV) Figures and Tables that are collapsible/expandable online. A maximum of 5 EV Figures can be typeset. EV Figures should be cited as 'Figure EV1, Figure EV2' etc... in the text and their respective legends should be included in the main text after the legends of regular figures.
 - For the figures that you do NOT wish to display as Expanded View figures, they should be bundled together with their legends in a single PDF file called *Appendix*, which should start with a short Table of Content. Appendix figures should be referred to in the main text as: "Appendix Figure S1, Appendix Figure S2" etc. See detailed instructions regarding expanded view here: <http://embor.embopress.org/authorguide#expandedview>.
 - Additional Tables/Datasets should be labeled and referred to as Table EV1, Dataset EV1, etc. Legends have to be provided in a separate tab in case of .xls files. Alternatively, the legend can be supplied as a separate text file (README) and zipped together with the Table/Dataset file.
- 7) We would also encourage you to include the source data for figure panels that show essential data.

Numerical data should be provided as individual .xls or .csv files (including a tab describing the data). For blots or microscopy, uncropped images should be submitted (using a zip archive if multiple images need to be supplied for one panel). Additional information on source data and instruction on how to label the files are available <http://embor.embopress.org/authorguide#sourcedata>.
- 8) Our journal encourages inclusion of *data citations in the reference list* to directly cite datasets that were re-used and obtained from public databases. Data citations in the article text are distinct

from normal bibliographical citations and should directly link to the database records from which the data can be accessed. In the main text, data citations are formatted as follows: "Data ref: Smith et al, 2001" or "Data ref: NCBI Sequence Read Archive PRJNA342805, 2017". In the Reference list, data citations must be labeled with "[DATASET]". A data reference must provide the database name, accession number/identifiers and a resolvable link to the landing page from which the data can be accessed at the end of the reference. Further instructions are available at <<http://embor.embopress.org/authorguide#datacitation>>.

9) Before submitting your revision, primary datasets (and computer code, where appropriate) produced in this study need to be deposited in an appropriate public database (see <<http://embor.embopress.org/authorguide#dataavailability>>).

The accession numbers and database should be listed in a formal "Data Availability " section (placed after Materials & Method) that follows the model below. Please note that the Data Availability Section is restricted to new primary data that are part of this study.

Data availability

10) Regarding data quantification, please ensure to specify the name of the statistical test used to generate error bars and P values, the number (n) of independent experiments underlying each data point (not replicate measures of one sample), and the test used to calculate p-values in each figure legend. Discussion of statistical methodology can be reported in the materials and methods section, but figure legends should contain a basic description of n, P and the test applied. Please note that error bars and statistical comparisons may only be applied to data obtained from at least three independent biological replicates. Please also include scale bars in all microscopy images.

I look forward to seeing a revised version of your manuscript when it is ready. Please let me know if you have questions or comments regarding the revision.

REFeree REPORTS

Referee #1:

Taulet and Vitre et al. provide evidence for a function of IFT (Intraflagellar Transport) proteins in centrosome clustering. Proteins of the IFT B complex, exemplified by IFT88 and IFT52, act together with minus end directed MT motor proteins, Dynein and HSET, to bring supernumerous centrosomes into one spindle pole favoring bipolar spindle assembly. The conclusion is based on a number of experiments, in which the authors quantify multipolar spindles or unclustered centrosomes in anaphase of non-transformed and cancer cells, directly track centrosomes in real time and look at the dynamics / MT binding behavior of the MT motor HSET using FRAP experiments. The authors compare control cells with knockdown or inhibitory (MT motors) conditions and exploit acute loss of function after induced protein degradation of IFT88 using the

AID system. Importantly, the authors provide evidence for a direct interaction of HSET (motor domain) and the IFT B complex consistent with a functional interaction in mitotic centrosome clustering.

Taken together, this is a very solid and experimentally sound study, which, in principle, deserves publication in EMBO reports with its current contents. Although differences measured are often rather small, all the experiments are carefully quantified and thus support the conclusions. They consistently underline the main take home message of IFT proteins working together with MT minus end directed motors in centrosome clustering.

Some issues, however, should be addressed prior to publication.

1. siRNA experiments are well quantified but only in one case further validated by a rescue experiment (Fig. 1D, IFT52). However, there is no information about expression of the rescue construct, i.e. what are expression levels in individual cells, how do levels of exogenous and endogenous protein before knockdown compare. Knockdown of IFT88 is parallel by AID experiments in DLD-1 cells but no validation (i.e. rescue) experiments are provided for any of the cancer cell lines investigated in Fig. 4 (e.g. Caco-2).
2. The interpretation of the FRAP experiment remains unclear to me. The authors talk about an altered "turnover" of the HSET motor after auxin-induced degradation of IFT88. To my opinion, the experiment only documents reduced efficiency rather than velocity of HSET rebinding after bleaching. Even that difference is rather small. This needs to be clarified: what exactly is the conclusion from this experiment: is it supposed to tell us that MT binding, processivity or velocity of MT motors is altered, or rather the pool of motors that can be recruited to spindle MT?
3. The function of centrosome clustering suffers even more in cancer cells (Fig. 4), which already display binucleation after IFT88 knockdown. However, the history of these cells is not well documented. When was knockdown started compared to live imaging? One would suppose that the binucleated cells already have supernumerous centrosomes due to cytokinesis failure. This should be tested and added to the data to support the conclusion.

Minor issues:

Fig. 1: Could the authors please remind us of the function of the p38 inhibitor used here. To indicate that siRNA was present only from -23 to -17 h is formally correct but oddly misleading as the knockdown conditions will remain even after removal of excess of transfection/siRNA reagents.

Fig. 2 C to F would benefit from defining the %age of material used also in the pull-down fraction (always 100%). The IFT88 signals in panel E are rather weak, why is that?

The scalebars in Figs EV1 A and B are 5 and 10 μ m while the condensed chromatin looks very comparable in size. Please double check.

A number of grammatical and semantic errors sometimes detract the reader from the contents of the actual experiments.

Referee #2:

Centrosome amplification, commonly observed in cancer cells, impose a mitotic stress to cells as they need to be able to cluster extra centrosomes to continue proliferating. Thus, it has been thought for many years that targeting the process of centrosome clustering could selectively kill cancer cells. Mechanistically, centrosome clustering requires different pathways, including cortical contractility, mitotic spindle fibers and microtubule motors, mainly dynein and HSET. The kinesin HSET is perhaps the main player as without HSET very few cells are able to cluster. However, how HSET activity is regulated and how the different pathways work together to enable efficient clustering is only now starting to be understood.

In this manuscript, Vitre et al propose a novel mechanism by which IFT proteins, previously implicated in regulating motors, could also facilitate HSET-mediated clustering. They found that some IFT proteins are important for centrosome clustering in normal and cancer cells with amplified centrosomes. Furthermore, they found that IFTs important for clustering associate with HSET in

in vitro and their depletion decrease HSET dynamics in the spindle, suggesting that they could play a role in regulating HSET activity. The authors demonstrate that depletion of IFTs lead to increase extra centrosomes distance after NEB that could compromise clustering, leading to increase multipolar mitoses. Finally, the authors provide evidence that depletion of IFTs in cells with extra centrosomes decreases their viability, further supporting a role for these proteins in centrosome clustering.

This manuscript provides the first evidence for a role for IFTs in the regulation of centrosome clustering and HSET activity. Overall, the experiments and data presented are convincing and of interest to the cell biology community. It is however unclear how mechanistically IFTs regulate HSET and there are few important controls missing. Therefore, the specific comments below should be addressed prior to publication.

#1. It would be helpful to add a graph with the centrosome amplification for all the Plk4-inducible cell lines used in this manuscript. The authors provide some of the % in the figure legends but is less clear. Also, both control (-dox) and Plk4-OE (+dox) % of extra centrosomes should be always provided.

#2. The authors should provide evidence that IFT siRNA does not lead to multipolar spindles/anaphases in -dox treated cells and provide images clearly demonstrating that IFT depletion leads to de-clustering and not centriole splitting, which is not very obvious from the images provided. All centrosomes/centrioles should be inside insets.

#3. Inhibition of HSET using CW069 inhibitor, used here as well, or HSET depletion has been shown to virtually lead to almost complete de-clustering of extra centrosomes. However, the authors found that only ~25% of multipolar anaphases in Fig 1I, which was surprising. It is difficult to know how much HSET is inhibited by CW069 in the conditions used. To address this, the authors should deplete HSET by siRNA in these cells and compare it with siRNA of IFTs. This is a better comparison, both proteins being depleted, plus levels of HSET depletion can be assessed by western blot.

#4. In Fig 2 the authors demonstrate motor domain and FL of HSET is able to interact with IFTs. However, these experiments are performed via pull-down and there is no evidence of such interaction in vivo. Can the authors perform immunoprecipitation of endogenous proteins to see if they interact inside cells? If this cannot be done, I would suggest tone down statements claiming interaction. Also, is there any reason why MDA-231 extracts were used in this experiment?

#5. In Fig 2G-H the authors show that loss of IFTs affect HSET dynamics by FRAP. Can the authors also measure tubulin levels in this experiment? From the images provided it seems that this could also be compromised?

#6. In Fig 3A its very difficult to know what are the centrin dots supposed to represent? Is it a pair of centrioles? Or 1 centriole?

#7. In Fig EV2 C,D, depletion of IFT88 also affects control -dox cells and 2N cells. While I understand that IFT have other functions in the spindle that could affect viability, it will be important to show how the spindles of these cells look like. Also, is there a reason why DLD-1 control cells are more affected than HCT-116 controls? Can the authors provide images for all colony assays performed? The 2 images images provided are not sufficient.

#8. Can the authors provide some explanation/discussion of how they think IFTs affect HSET activity? If they bind to the motor domain shouldn't their presence negatively impact HSET motor activity? Could the authors discuss what they think is happening?

#9. Previous work has shown that the distance between centrosomes is important to facilitate HSET-mediated clustering. The authors should discuss these findings (Rhys et al, 2018) in light of their results.

Referee #3:

Centrosome amplification is a common feature of cancer cells and is thought to be a contributing factor in tumorigenesis. During mitotic cell division, supernumerary centrosomes increase the likelihood of chromosome missegregation and aneuploidy. However, most cells still assemble a bipolar spindle thanks to robust centrosome clustering mechanisms involving the kinesin HSET/KIFC1 as well as cytoplasmic dynein. Since tumor cells would otherwise undergo a lethal multipolar division, a better understanding of the molecular mechanisms underlying centrosome clustering is clearly of great scientific and clinical interest.

Here, Vitre and colleagues describe a role for components of the intraflagellar transport (IFT) machinery, specifically the IFT-B core complex of IFT88,70,52 and 46, in centrosome clustering independent of their role in cilium assembly. Using RNAi and auxin inducible degradation, the authors show that these IFT proteins are required for bipolar division in cells where centrosome amplification has been induced by overexpressing the driver of centriole assembly PLK4. The authors propose that IFT proteins act in the same pathway as HSET and dynein since their simultaneous inhibition/ depletion does not result in a synergistic effect. Consistent with this the authors report a direct protein-protein interaction between HSET and the IFT-B core complex. Finally, the authors extend their findings to cancer cell lines in which centrosome amplification naturally occurs.

On the whole the experiments in this manuscript are well executed and the findings novel and of high potential significance. As such, this manuscript certainly meets the standards for EMBO Reports. However, as detailed below there are a few points that need to be addressed prior to publication. Further, the manuscript at present is not well written and would benefit from a thorough rewrite and potentially professional editing to make its findings more accessible to the reader.

Major points

1. As the authors stress the minus-end-directed kinesin HSET/KIFC1 is of particular interest since it is essential for centrosome clustering but not for the division of cells with two centrosomes (page 3). Given that IFT components have been reported to contribute to other aspects of mitosis, it would be good to show to what extent they are likewise dispensable for spindle bipolarity in the absence of centrosome amplification.

2. The authors make much of the lack of a synergistic phenotype when co-inhibiting/depleting HSET/dynein and IFT-B components. They further argue that IFT-B proteins form a complex that interacts with HSET and regulates its motor activity. A clear prediction, then, is that co-depletion of multiple IFT-B components likewise results in no synergistic effects. This should be tested experimentally.

3. Figure 3D appears to monitor centrosome cohesion rather than clustering. A better way of monitoring defects in centrosome clustering would be a quantitation of the number of tripolar mitoses being resolved by anaphase onset.

4. I found this manuscript quite difficult to read due to poor language and idiosyncratic presentation. Below I am highlighting just a few instances:

Manuscript text

- 'Centrosomes clustering' (title and throughout manuscript) should read: 'centrosome clustering'

- 'IFTs' (title and throughout manuscript) should read 'IFT proteins'

- 'In dividing cells, centrosomes allow for proper assembly of a functional bipolar mitotic spindle' (page 3) should read 'contribute to'

- 'Multiple works' (page 3) should read: 'Multiple studies'

- 'participating to proper chromosomes alignment' (page 3) should read: 'contributing to proper chromosome alignment'

- 'central spindle microtubules architecture' (page 4) should read: 'microtubule architecture'

- 'Similarities between this later mechanism' (page 4) should read: 'latter mechanism'

- 'cells doing multipolar anaphases' (page 5) should read: 'undergoing'

- 'This was confirmed using only purified proteins with both FL and motor GFP-HSET pulling-down purified IFT88/70/52/46 complex while tail GFP-HSET was not (Fig 2F).' (page 7) 'Was not' should read: 'did not'

- 'we observed a 20% increase in this distance for siCT versus si52 conditions (3,1 μm versus 3,9

μm) and a 24% increase between DMSO and CW069 conditions (3,2 μm versus 4,2 μm). (page 9). si52 and siCT and CW069 and DMSO need to be switched around for this sentence to make sense.

Figures

- A simplified schematic of IFT proteins and in particular the B core complex would be helpful to guide readers from the mitosis field unfamiliar with IFT.
- The localization of IFT52 and IFT88 to mitotic spindle poles and for IFT88 demonstrating the specificity of this signal is a major finding and should not be hidden away in a Supplemental Figure (Figure EV1A,B,H).
- In contrast, Figure 3 at present does not convey any important information and could be moved to the Supplement.
- Figure EV2A. The DMSO condition in the graph serves no purpose at all and should be removed.
- The presentation of the pulldown data (Figure 2) is somewhat unusual. Why blot beads prior to incubation with extract (B, input)?

Other points

5. It would be good to show/quantitate the level of centrosome amplification achieved by overexpression of PLK4.
6. For the quantitation in Figure 2H, it would be good to show the starting levels and distribution of HSET on the spindle since this could influence the recovery rate. At present all conditions are normalized to 100% pre-bleach.
7. 'We found that IFT52 depletion as well as HSET inhibition increased the average duration of the first period compared to control conditions (106 min in siCT vs 165 in si52 and 141 min in DMSO vs 178 min in CW069; Fig 3B).' (page 8) This is not so. According to panel 3B the difference between control and HSET inhibition is not statistically significant.
8. The pull down of IFT88 in Figure 2E is not convincing at all and would need to be repeated. However, since the same mapping of HSET/IFT88 interaction is shown with purified components in Figure 2F this panel could also be removed.
9. While this may perhaps be beyond the scope of the current paper, what the authors cannot strictly exclude at present is indirect effects due to impaired cilia-dependent signaling following IFT perturbation. This could be done eg by RNAi or knockout of a centriolar or transition zone component specifically required for ciliogenesis.

1st Revision - authors' response

24 January 2020

Referee #1:

Taulet and Vitre et al. provide evidence for a function of IFT (Intraflagellar Transport) proteins in centrosome clustering. Proteins of the IFT B complex, exemplified by IFT88 and IFT52, act together with minus end directed MT motor proteins, Dynein and HSET, to bring supernumerous centrosomes into one spindle pole favoring bipolar spindle assembly. The conclusion is based on a number of experiments, in which the authors quantify multipolar spindles or unclustered centrosomes in anaphase of non-transformed and cancer cells, directly track centrosomes in real time and look at the dynamics / MT binding behavior of the MT motor HSET using FRAP experiments. The authors compare control cells with knockdown or inhibitory (MT motors) conditions and exploit acute loss of function after induced protein degradation of IFT88 using the AID system. Importantly, the authors provide evidence for a direct interaction of HSET (motor domain) and the IFT B complex consistent with a functional interaction in mitotic centrosome clustering.

Taken together, this is a very solid and experimentally sound study, which, in principle, deserves publication in EMBO reports with its current contents. Although differences measured are often

rather small, all the experiments are carefully quantified and thus support the conclusions. They consistently underline the main take home message of IFT proteins working together with MT minus end directed motors in centrosome clustering.

Some issues, however, should be addressed prior to publication.

1. siRNA experiments are well quantified but only in one case further validated by a rescue experiment (Fig. 1D, IFT52). However, there is no information about expression of the rescue construct, i.e. what are expression levels in individual cells, how do levels of exogenous and endogenous protein before knockdown compare. Knockdown of IFT88 is parallel by AID experiments in DLD-1 cells but no validation (i.e. rescue) experiments are provided for any of the cancer cell lines investigated in Fig. 4 (e.g. Caco-2).

Re: We are now providing a western blot illustrating the level of exogenous mouse mcherryIFT52 expressed in the RPE-1 cells in the rescue experiments (Fig EV1C). We have also carried out a rescue experiment in the Caco2 cell line as requested. Because overexpression of IFT88 is toxic for the cells, we have performed siRNA depletion and rescue of IFT52. siRNA of IFT52 in Caco-2 is not highly effective as illustrated in Fig EV3C. Nevertheless, this depletion is sufficient to triggers a significant increase in un-clustered mitotic figures which can be rescued by re expressing exogenous mIFT52 in the Caco-2 cells. These new results are now presented in Fig 5E.

2. The interpretation of the FRAP experiment remains unclear to me. The authors talk about an altered "turnover" of the HSET motor after auxin-induced degradation of IFT88. To my opinion, the experiment only documents reduced efficiency rather than velocity of HSET rebinding after bleaching. Even that difference is rather small. This needs to be clarified: what exactly is the conclusion from this experiment: is it supposed to tell us that MT binding, processivity or velocity of MT motors is altered, or rather the pool of motors that can be recruited to spindle MT?

Re: From the analysis of the FRAP experiment we can only conclude that the pool of GFP-HSET motor that can be recruited on microtubules of the mitotic spindle is reduced following photobleaching. We have now edited the text to clarify that the depletion of HSET reduces the recruitment of HSET on the mitotic spindle following photobleaching. Understanding the exact molecular mechanisms underlying this reduced recruitment will require a dedicated biochemical/ in vitro works.

3. The function of centrosome clustering suffers even more in cancer cells (Fig. 4), which already display binucleation after IFT88 knockdown. However, the history of these cells is not well documented. When was knockdown started compared to live imaging? One would suppose that the binucleated cells already have supernumerous centrosomes due to cytokinesis failure. This should be tested and added to the data to support the conclusion.

Re: We started the knockdown of IFT88 in the MDA-MB-231 23 hours before the beginning of the live imaging. We have now added a schematic to describe the timing of the experiment with the binucleated MDA-MB-231 cells (Fig 5G). The binucleated MDA-MB-231 cells are naturally present within the global population of MDA-MB-231 cells we are imaging. As the reviewer suggests this binucleation is most likely the consequence of cytokinesis failure. We have measured the proportion of centrosome amplification by immunofluorescence in the binucleated cells and, as expected, found that 67,1 % of those cells present centrosome amplification compared to 27,1% of the total MDA-MB-231 cell population presenting centrosome amplification. We have now added this quantification to Fig 5A, we have also edited the text to indicate the proportion of centrosome amplification and the possible origin of this amplification in the MDA-MB-231 cells.

Minor issues:

Fig. 1: Could the authors please remind us of the function of the p38 inhibitor used here.

Re: We use the p38 stress-activated protein kinase inhibitor to allow the RPE-1 cells to have a continuous proliferation in the presence of extra centrosomes. Indeed, Plk4 overexpression and its related centrosome amplification trigger p38 activation and cell cycle arrest in RPE-1 cells as documented in Holland et al, *Genes and Development*, 2012. The justification for the use of p38 inhibitor is now explained in the text (methods).

To indicate that siRNA was present only from -23 to -17 h is formally correct but oddly misleading as the knockdown conditions will remain even after removal of excess of transfection/siRNA reagents.

Re: We have corrected the diagram, now in Fig 1B, according to reviewer's suggestion.

Fig. 2 C to F would benefit from defining the %age of material used also in the pull-down fraction (always 100%).

Re: We are now providing the percentages of material of both input and pull-down for all the pull-down experiments

The IFT88 signals in panel E are rather weak, why is that?

We were not able to recapitulate the strong signal obtained in former Fig 2C for pull-down of IFT88 with FL GFP HSET on former Fig 2E with the similar FL GFP-HSET. This could come from the different devices used for the western blots. In Fig 2C we used a Biorad mini protean 3 system, while in the Fig 2E we used a Biorad Criterion Blotter system and precast gels. We usually have dimer signal with the Criterion system.

Reviewer 3 (point #8), has also questioned the dim signal presented in former Fig 2E and suggested to remove this panel as former Fig 2F present similar data. We have now removed former Fig 2E, kept former Fig 2F (now Fig 3G) and performed new experiments of pull-down and blotting of various IFT proteins with motor GFP HSET to show the absence of binding of IFT27 and IFT140 to this construct. These results are now presented in a new panel (Fig 3H).

The scalebars in Figs EV1 A and B are 5 and 10 um while the condensed chromatin looks very comparable in size. Please double check.

Re: Thank you for picking this mistake, the scale was indeed wrong in former Fig EV1B, it is now corrected in new Fig EV1B.

A number of grammatical and semantic errors sometimes detract the reader from the contents of the actual experiments.

Re: We have now requested assistance to correct grammatical and semantic errors. We hope that the reviewer will find the manuscript better written in this new version.

Referee #2:

Centrosome amplification, commonly observed in cancer cells, impose a mitotic stress to cells as they need to be able to cluster extra centrosomes to continue proliferating. Thus, it has been thought for many years that targeting the process of centrosome clustering could selectively kill cancer cells. Mechanistically, centrosome clustering requires different pathways, including cortical contractility, mitotic spindle fibers and microtubule motors, mainly dynein and HSET. The kinesin HSET is perhaps the main player as without HSET very few cells are able to cluster. However, how HSET activity is regulated and how the different pathways work together to enable efficient clustering is only now starting to be understood.

In this manuscript, Vitre et al propose a novel mechanism by which IFT proteins, previously implicated in regulating motors, could also facilitate HSET-mediated clustering. They found that

some IFT proteins are important for centrosome clustering in normal and cancer cells with amplified centrosomes. Furthermore, they found that IFTs important for clustering associate with HSET in vitro and their depletion decrease HSET dynamics in the spindle, suggesting that they could play a role in regulating HSET activity. The authors demonstrate that depletion of IFTs lead to increase extra centrosomes distance after NEB that could compromise clustering, leading to increase multipolar mitoses. Finally, the authors provide evidence that depletion of IFTs in cells with extra centrosomes decreases their viability, further supporting a role for these proteins in centrosome clustering.

This manuscript provides the first evidence for a role for IFTs in the regulation of centrosome clustering and HSET activity. Overall, the experiments and data presented are convincing and of interest to the cell biology community. It is however unclear how mechanistically IFTs regulate HSET and there are few important controls missing. Therefore, the specific comments below should be addressed prior to publication.

#1. It would be helpful to add a graph with the centrosome amplification for all the Plk4-inducible cell lines used in this manuscript. The authors provide some of the % in the figure legends but is less clear. Also, both control (-dox) and Plk4-OE (+dox) % of extra centrosomes should be always provided.

Re: As requested, we have now added a diagram (Fig EV1A) summarizing the levels of centrosome amplification for all the tetracyclin/doxycycline inducible cell lines used in the study. The percentage of centrosome amplification for diploid and tetraploid HCT-116 is also presented on the diagram.

#2. The authors should provide evidence that IFT siRNA does not lead to multipolar spindles/anaphases in -dox treated cells and provide images clearly demonstrating that IFT depletion leads to de-clustering and not centriole splitting, which is not very obvious from the images provided. All centrosomes/centrioles should be inside insets.

Re: We have now quantified the proportion of multipolar spindles/anaphases in RPE-1 and DLD-1 cells that do not present extra centrosomes (-dox). This proportion is below 5% in all cases. Moreover, there are no significant differences between control and perturbed conditions (Fig EV1I and Fig EV2G). This indicates that IFT proteins depletion does not trigger multipolar mitotic figures in absence of extra centrosomes. We have also added images of RPE-1 and DLD-1 cells without centrosome amplification and IFT perturbation to illustrate this observation (Fig EV1H and Fig EV2A-B). We agree that the cell highlighted in Figure 1L presents one spindle pole where only one centrin dot is present. This is misleading as all the other cells with multipolar spindles present in the field of view have two centrin dots per pole. Therefore, we have changed the highlighted cell, in the same field of view (new Figure 2B). We also provide additional images of multipolar RPE-1 and DLD-1 cells with at least two centrin dots per spindle pole (FigEV1H, FigEV1K, FigEV2A-B). Finally, we have quantified the proportion of spindle poles with less than two centrin dots in the RPE-1 and DLD-1 cells. This proportion is below 10% in the two cell lines and there are no significant changes in those proportions between controlled and perturbed conditions (Fig EV1J). This result indicates that depletion of IFT88 or IFT52 does not trigger centriole splitting.

#3. Inhibition of HSET using CW069 inhibitor, used here as well, or HSET depletion has been shown to virtually lead to almost complete de-clustering of extra centrosomes. However, the authors found that only ~25% of multipolar anaphases in Fig 1I, which was surprising. It is difficult to know how much HSET is inhibited by CW069 in the conditions used. To address this, the authors should deplete HSET by siRNA in these cells and compare it with siRNA of IFTs. This is a better comparison, both proteins being depleted, plus levels of HSET depletion can be assessed by western blot.

Re: For the experiments with the small molecule inhibitors of the motor, we voluntarily used concentrations of inhibitor that recapitulate the level of perturbation achieved with IFT proteins siRNA. Our goal was to avoid masking potential cumulative defects in the co-perturbations assay due to one perturbation being much stronger than the other. As requested by the reviewer, we have now quantified the proportion of multipolar anaphases observed upon HSET depletion by siRNA. We observed 59 % of multipolar anaphases upon HSET depletion compared to 9,3 % in the control condition in this set of experiments (Fig EV2H-I). So, indeed, the complete depletion of HSET leads to a stronger effect than the inhibition of HSET or the depletion of IFT52. We need to point out here that IFT52 higher/complete depletion could not be achieved even with higher concentration of siRNA or using different transfection methods. As expected when we do the co-depletion of HSET and IFT52 by siRNA, we don't see significant changes in the proportion of multipolar anaphases as compared to single HSET depletion.

#4. In Fig 2 the authors demonstrate motor domain and FL of HSET is able to interact with IFTs. However, these experiments are performed via pull-down and there is no evidence of such interaction *in vivo*. Can the authors perform immunoprecipitation of endogenous proteins to see if they interact inside cells? If this cannot be done, I would suggest tone down statements claiming interaction. Also, is there any reason why MDA-231 extracts were used in this experiment?

Re: We have now performed and included in Figure 3 immunoprecipitation of endogenous HSET and we show that IFT88 is co-immunoprecipitated with HSET indicating an interaction between the endogenous proteins in cells (Fig 3B). We used MDA-MB-231 cells because they naturally present centrosome amplification and clustering capacity. We also used those cells because they are easy to synchronize in mitosis (using nocodazole) and to collect by (mitotic) shake-off.

#5. In Fig 2G-H the authors show that loss of IFTs affect HSET dynamics by FRAP. Can the authors also measure tubulin levels in this experiment? From the images provided it seems that this could also be compromised?

Re: As requested, we quantified the tubulin intensity level upon IFT perturbation in the FRAP experiments. There are no differences in tubulin intensity levels between the different experimental conditions as shown in rebuttal Figure 1 below.

Rebuttal Figure 1: Level of tubulin signal intensity during FRAP experiment. Quantification of the fluorescence signal of the tubulin (SiR-tubulin) in DLD-1 cells during the FRAP experiments, following the indicated treatments. Mean + SEM of four independent experiments.

#6. In Fig 3A its very difficult to know what are the centrin dots supposed to represent? Is it a pair of centrioles? Or 1 centriole?

Re: The centrin dots observed *in vivo* in the still from live imaging in former Fig 3A (now figure 4A) represent a pair of centrioles but they cannot be resolved with a 40x objective. To address this concern, we have acquired successive z stacks of live mitotic cell with the 40x and the 100x objectives. With the 100x objective, the two centrin dots can be distinguished, despite one dot being usually much dimer than the other. We have now mentioned this in the text and added a related figure (Fig EV3A). In our live experiment we acquired the images with 40x objective as we needed to detect individual centrosomes, and the resolution of this objective is sufficient for this detection.

In addition, the use of a 40x objective allows us to obtain a sufficient number of mitotic events per experimental condition, during an overnight imaging. Thus, with a 2 minutes time interval and four imaging fields per condition we obtained, in average, 10 mitotic events that can be analyzed per condition per experiment.

#7. In Fig EV2 C,D, depletion of IFT88 also affects control -dox cells and 2N cells. While I understand that IFT have other functions in the spindle that could affect viability, it will be important to show how the spindles of these cells look like. Also, is there a reason why DLD-1 control cells are more affected than HCT-116 controls? Can the authors provide images for all colony assays performed? The 2 images images provided are not sufficient.

Re: We now provide images of the -dox DLD-1 and control HCT-116 mitotic spindles with or without IFT88 depletion (Fig EV2B and EV2D). No obvious global defects of the mitotic spindle were observed in the IFT88 depletion condition (see also #2). DLD-1 control cells (with dox) are more affected than HCT-116 control (4N) upon siCT because they have acute accumulation of extra centrosomes while HCT-116 are established tetraploid cell lines that had time to adapt to the presence of extra centrosomes. In other words, HCT-116 are only subjected to a control siRNA treatment while DLD-1 cells have to deal with the additional stress generated by the presence of newly generated supernumerary centrosomes. As requested, we now include images for all the conditions tested in the colony assays (Fig EV3F and EV3H).

#8. Can the authors provide some explanation/discussion of how they think IFTs affect HSET activity? If they bind to the motor domain shouldn't their presence negatively impact HSET motor activity? Could the authors discuss what they think is happening?

Re: We have now added an additional paragraph in the discussion regarding this aspect of HSET regulation by IFT proteins: *“Kinesins can be autoinhibited due to interaction between their motor domain and their tail domain [46]. This inhibition can be released through the binding of partner protein as shown for kinesin 1 motor[47]. Conversely, Kinesin 14 family member KCPB, a plant kinesin related to HSET, is known to be inhibited by a regulating partner proteins KIC [48]. Considering those regulatory mechanisms of kinesins by partner proteins, it is possible that IFT proteins can activate HSET by competing with inhibiting regulator of HSET motor or through the release of an autoinhibition. This latter mechanism is strongly supported by a recent study that shows the activation of kinesin 2 by IFT-B subcomplex proteins, made of IFT46-IFT52-IFT70-IFT88, in vitro [36].”*

#9. Previous work has shown that the distance between centrosomes is important to facilitate HSET-mediated clustering. The authors should discuss these findings (Rhys et al, 2018) in light of their results.

Re: We have now added an additional paragraph in the discussion regarding HSET mediated clustering and the distance between centrosomes and compared our results to the ones presented in Rhys et al, 2018 : *“Interestingly the range of distance measured between centrosomes that are clustered at anaphase onset (from 5,6 μm at NEB to 3,1μm at anaphase onset) is within the range of distance where the kinesin HSET is acting through its motor activity to promote centrosome clustering, as recently shown in MCF10a cells [19]”.*

Referee #3:

Centrosome amplification is a common feature of cancer cells and is thought to be a contributing factor in tumorigenesis. During mitotic cell division, supernumerary centrosomes increase the likelihood of chromosome missegregation and aneuploidy. However, most cells still assemble a bipolar spindle thanks to robust centrosome clustering mechanisms involving the kinesin HSET/KIFC1 as well as cytoplasmic dynein. Since tumor cells would otherwise undergo a lethal

multipolar division, a better understanding of the molecular mechanisms underlying centrosome clustering is clearly of great scientific and clinical interest.

Here, Vitre and colleagues describe a role for components of the intraflagellar transport (IFT) machinery, specifically the IFT-B core complex of IFT88,70,52 and 46, in centrosome clustering independent of their role in cilium assembly. Using RNAi and auxin inducible degradation, the authors show that these IFT proteins are required for bipolar division in cells where centrosome amplification has been induced by overexpressing the driver of centriole assembly PLK4. The authors propose that IFT proteins act in the same pathway as HSET and dynein since their simultaneous inhibition/ depletion does not result in a synergistic effect. Consistent with this the authors report a direct protein-protein interaction between HSET and the IFT-B core complex. Finally, the authors extend their findings to cancer cell lines in which centrosome amplification naturally occurs.

On the whole the experiments in this manuscript are well executed and the findings novel and of high potential significance. As such, this manuscript certainly meets the standards for EMBO Reports. However, as detailed below there are a few points that need to be addressed prior to publication. Further, the manuscript at present is not well written and would benefit from a thorough rewrite and potentially professional editing to make its findings more accessible to the reader.

Major points

1. As the authors stress the minus-end-directed kinesin HSET/KIFC1 is of particular interest since it is essential for centrosome clustering but not for the division of cells with two centrosomes (page 3). Given that IFT components have been reported to contribute to other aspects of mitosis, it would be good to show to what extent they are likewise dispensable for spindle bipolarity in the absence of centrosome amplification.

Re: We have now quantified and included in the figures the proportion of multipolar spindles/anaphases in RPE-1 and DLD-1 cells that do not present extra centrosomes (-dox). This proportion is below 5% in all cases. Moreover, there are no significant differences between control and perturbed conditions (Fig EV1I and Fig EV2G). This indicates that IFT proteins depletion does not trigger multipolar mitotic figures in absence of extra centrosomes. We also included images of RPE-1, DLD-1 and HCT-116 cells without centrosome amplification and IFT perturbation to illustrate this point (Fig EV1H and Fig EV2A-B and EV2D).

2. The authors make much of the lack of a synergistic phenotype when co-inhibiting/depleting HSET/dynein and IFT-B components. They further argue that IFT-B proteins form a complex that interacts with HSET and regulates its motor activity. A clear prediction, then, is that co-depletion of multiple IFT-B components likewise results in no synergistic effects. This should be tested experimentally.

Re: We agree with the referee regarding the expected results of IFT52 and IFT88 co-depletion. Indeed, we have now tested the co-depletion of the two IFT proteins and, as expected, we observed no cumulative defects in the proportion of multipolar anaphases (Fig EV1F-G).

3. Figure 3D appears to monitor centrosome cohesion rather than clustering. A better way of monitoring defects in centrosome clustering would be a quantitation of the number of tripolar mitoses being resolved by anaphase onset.

Re: The centrin dots observed *in vivo* in the still from live imaging in former Fig 3A (now figure 4A) represent a pair of centrioles but they cannot be resolved with a 40x objective. We have acquired successive images of live mitotic cell with the 40x and the 100x objective. With the 100x objective, the two centrin dots can be distinguished, despite one dot being usually much dimmer than the other. We have now mentioned this in the text and added a related figure (Fig EV3A). Thus, in

the new Figure 4, we are indeed monitoring centrosome clustering and not centriole cohesion. In addition, we have now assessed centriole cohesion. More than 90 % of spindle poles have at least two centrin dots (centrioles) and this remains unchanged between control and perturbed conditions in the RPE-1 cells with centrosome amplification (Fig EV1J-K). The cells used in new Figure 4 are the same cells that are used in Figure 1 so we know from Figure 1 that IFT proteins depletion leads to an increase in multipolar mitosis that are not resolved at anaphase onset (Fig 1E). The aim of Figure 4, and in particular Fig 4D is to measure the cohesion between centrosomes upon control and perturbed conditions and to show that IFT depletion or HSET inhibition results in a reduction in centrosome cohesion and an increase distance between adjacent centrosomes (Fig 4D, black arrow).

4. I found this manuscript quite difficult to read due to poor language and idiosyncratic presentation. Below I am highlighting just a few instances:

Manuscript text

- 'Centrosomes clustering' (title and throughout manuscript) should read: 'centrosome clustering'

- 'IFTs' (title and throughout manuscript) should read 'IFT proteins'

- 'In dividing cells, centrosomes allow for proper assembly of a functional bipolar mitotic spindle' (page 3) should read 'contribute to'

- 'Multiple works' (page 3) should read: 'Multiple studies'

- 'participating to proper chromosomes alignment' (page 3) should read: 'contributing to proper chromosome alignment'

- 'central spindle microtubules architecture' (page 4) should read: 'microtubule architecture'

- 'Similarities between this later mechanism' (page 4) should read: 'latter mechanism'

- 'cells doing multipolar anaphases' (page 5) should read: 'undergoing'

- 'This was confirmed using only purified proteins with both FL and motor GFP-HSET pulling-down purified IFT88/70/52/46 complex while tail GFP-HSET was not (Fig 2F).' (page 7) 'Was not' should read: 'did not'

- 'we observed a 20% increase in this distance for siCT versus si52 conditions (3,1 μm versus 3,9 μm) and a 24% increase between DMSO and CW069 conditions (3,2 μm versus 4,2 μm).' (page 9). si52 and siCT and CW069 and DMSO need to be switched around for this sentence to make sense.

Re: We are sorry for the multiple errors the referee found in the original manuscript. We have now requested assistance to correct grammatical and semantic errors. We hope that the reviewer will find the manuscript better written in this new version.

Figures

- A simplified schematic of IFT proteins and in particular the B core complex would be helpful to guide readers from the mitosis field unfamiliar with IFT.

Re: We have now added a schematic of IFT proteins from the subcomplex B in new Figure 3A.

- The localization of IFT52 and IFT88 to mitotic spindle poles and for IFT88 demonstrating the specificity of this signal is a major finding and should not be hidden away in a Supplemental Figure (Figure EV1A,B,H).

Re: We have moved to Figure 1 an image that shows the localization of IFT52 to the mitotic spindle poles to (Fig 1A). We kept the image illustrating IFT88 localization to the spindle pole in the new Fig EV1B as this localization is already described and published in other works (Robert et al, J. Cell Sci, 2007; Delaval et al, Nat. Cell Biol., 2011; Taulet et al, Nat. Comm., 2017).

- In contrast, Figure 3 at present does not convey any important information and could be moved to the Supplement.

Re: As explained in details in response to point 3 of the referee, we think former Figure 3 (new Figure 4) conveys important information as it provides new insights on the cohesion of centrosome

during mitosis. It also provides new information regarding the dynamics (and the cohesion) of centrosomes in late G2 and in mitosis. Therefore, we would like to keep this figure as a main figure.

- Figure EV2A. The DMSO condition in the graph serves no purpose at all and should be removed.

Re: We have removed the DMSO condition from the diagram displayed in new Fig EV3B.

- The presentation of the pulldown data (Figure 2) is somewhat unusual. Why blot beads prior to incubation with extract (B, input)?

We displayed the blot of the beads without GFP-HSET, prior to incubation extract (B, input) to show that the GFP trap beads generate no signal by themselves at the molecular weight where we were looking for pull-downed proteins. We agree that displaying this condition is not required and we removed them from the blots in new Figure 3E-F.

Other points

5. It would be good to show/quantitate the level of centrosome amplification achieved by overexpression of PLK4.

Re: We have now added a diagram (Fig EV1A) that recapitulates the levels of centrosome amplification for all the tetracyclin/doxycycline inducible cell lines used in the study. The percentage of centrosome amplification for diploid and tetraploid HCT-116 is also present on diagram.

6. For the quantitation in Figure 2H, it would be good to show the starting levels and distribution of HSET on the spindle since this could influence the recovery rate. At present all conditions are normalized to 100% pre-bleach.

Re: The raw (not normalized) measurements of HSET intensity on the spindle at the start of the experiment are now provided in the Rebuttal Figure 2 Below. This figure shows the raw level and the distribution of HSET intensity at the last time point before photobleaching (Time 0). There are no significant differences in the level of HSET between control and perturbed conditions. This quantification can be included in the manuscript upon referee's request.

Rebuttal Figure 2: Levels of HSET before photobleaching. Quantification of the fluorescence signal intensity of the GFP HSET in DLD-1 cells, at the last time point before photobleaching for the indicated conditions. The horizontal line is the mean intensity of four independent experiments. ns: not significant. Unpaired t-test. A minimum of 41 measurements per condition were tested.

7. 'We found that IFT52 depletion as well as HSET inhibition increased the average duration of the first period compared to control conditions (106 min in siCT vs 165 in si52 and 141 min in DMSO vs 178 min in CW069; Fig 3B).' (page 8) This is not so. According to panel 3B the difference between control and HSET inhibition is not statistically significant.

Re: We agree with the referee that the increase in the average duration for HSET inhibition is not significantly higher. We have modified the text to make this clear.

8. The pull down of IFT88 in Figure 2E is not convincing at all and would need to be repeated.

However, since the same mapping of HSET/IFT88 interaction is shown with purified components in Figure 2F this panel could also be removed.

Re: We have now removed former Fig 2E and kept former Fig 2F (now Fig 3G). New experiments were performed for pull-down and blotting of various IFT proteins with motor GFP HSET to show the absence of binding of IFT27 and IFT140 to this construct in a new panel (Fig 3H).

9. While this may perhaps be beyond the scope of the current paper, what the authors cannot strictly exclude at present is indirect effects due to impaired cilia-dependent signaling following IFT perturbation. This could be done eg by RNAi or knockout of a centriolar or transition zone component specifically required for ciliogenesis.

Re: During the course of our experiments, we were looking at cycling cells that do not grow cilia so depletion of IFT proteins in our experiments should not have generated cilia dependent perturbation. Importantly, this possibility can be excluded at least for the DLD-1 cells. First because the DLD-1 cells do not grow cilia (Lancaster et al, Nat. Cell Biol., 2011). Second, because in the experiment presented in Figure 2B-F the cells are arrested in mitosis for 2 hours and are acutely depleted of IFT88 in mitosis, using the AID system. So, in this experiment, IFT88 depletion only occurs in mitosis where no cilia related mechanisms are involved because of the absence of cilia in mitosis. Those observations, and the fact that we report similar defects in DLD-1 and RPE-1 cells upon IFT proteins perturbation led us to conclude that the defects we observed in centrosome clustering upon IFT proteins depletion are independent of a potential impaired-cilia-dependent signaling.

2nd Editorial Decision

14 February 2020

Thank you for submitting the revised version of your manuscript. It has now been seen by all of the original referees.

As you can see, the referees find that the study is significantly improved during revision and recommend publication. Before I can accept the manuscript, I need you to address some minor points below:

- Please address the remaining minor concerns of referee #3.
- Please provide 3-5 keywords for your study. These will be visible in the html version of the paper and on PubMed and will help increase the discoverability of your work.
- We noted that there are currently two AD's in the Author Contributions section. For clarity purposes one should be ADou and the other ADos.
- Movies should be zipped with their legends and legends should be removed from the manuscript text.
- The following need scale bars: Fig 1A+F (the magnified boxes), Fig 2C+D (the magnified boxes), Fig 3I (the magnified boxes), Fig 5B (the magnified boxes), Fig EV1B+D (the magnified boxes), Fig EV2A,B+D (the magnified boxes), Fig EV3A+D (the magnified boxes)
- We realized that the upper IFT88 and tubulin blots of Figure 3C look overcontrasted, whereas the source data look much better. Please consider replacing these panels with the source data.
- Papers published in EMBO Reports include a 'Synopsis' to further enhance discoverability. Synopses are displayed on the html version of the paper and are freely accessible to all readers. The synopsis includes a short standfirst summarizing the study in 1 or 2 sentences that summarize the key findings of the paper and are provided by the authors and streamlined by the handling editor. I would therefore ask you to include your synopsis blurb.
- In addition, please provide an image for the synopsis. This image should provide a rapid overview of the question addressed in the study but still needs to be kept fairly modest since the image size cannot exceed 550x400 pixels.
- Our production/data editors have asked you to clarify several points in the figure legends (see attached document). Please incorporate these changes in the attached word document and return it with track changes activated.

Thank you again for giving us to consider your manuscript for EMBO Reports, I look forward to

your minor revision.

REFeree REPORTS

Referee #1:

Vitre and Taulet hand in a revised version of their ms on IFT proteins and HSET to promote extracentrosome clustering in mitosis.

The new version closes the few little gaps that were left in the initial submission. The authors provide additional, well-documented and quantified rescue experiments for their IFT52 knockdowns in RPE-1 and Caco2 cells. They correlate binucleation penetrance with centrosome amplification in cancer cells and add a graphical overview to document the timing of these knock-down experiments. Additional experimental data and quantifications match suggestions of reviewers 2 and 3. The authors further carefully address formal issues that were raised in my initial review, including the discussion of the presented FRAP experiments as well as a general correction of wording and grammar.

I am, therefore, in favor of publishing the manuscript in its present form in EMBO reports.

Referee #2:

The authors address my initial concerns/comments and thus i would support its publication as it is to EMBO Reports.

The addition of insets where centrioles are visible is very helpful. However, these images are very pixelated. Can the authors add better quality insets?

Referee #3:

The revised manuscript by Vitre et al largely addresses the points made in my original review. I am therefore in favor of publication in EMBO Reports. However, a few misstatements and inaccuracies persist or have been introduced in the revision as detailed below. I strongly recommend that the authors consider the following points in preparing their final manuscript draft:

1. In their discussion of the tracking experiment presented in Figure 4 (now clarified to be a centrosome, not centriole, tracking experiment, to my original point 3), the authors make the following statement: 'Those data indicate that HSET activity on centrosome dynamics starts before NEB and is mediated by IFT52' (page 10). I am not sure from what data the authors infer such a dependency relationship. While the phenotype the authors observe following IFT52 depletion may be stronger than that following HSET inhibition, this does not place HSET upstream (or downstream) of IFT52.
2. To my original point 4, I appreciate the authors efforts to improve the manuscript text. To me it still reads incredibly dense and occasionally confusing, but still an improvement on the original. One point the authors may still wish to consider is whether the 'extra' in extra centrosome clustering is strictly necessary. In the title it may be worth stating 'IFT proteins interact with HSET to promote clustering of supernumerary centrosomes in mitosis'. Elsewhere in the text the 'extra' can be taken as understood.
3. Addressing my original point 5, Figure EV1A now presents a quantification of centrosome amplification after PLK4 induction. What is left unclear is how centrosome number was measured and what the authors consider as abnormal, ie >4 centrin foci or >2 gamma-tubulin?
4. Rebuttal Figure 2 shows that the levels of HSET on the mitotic spindle are unaffected by inhibition with CW069 or depletion of IFT88, addressing part of my original point 6 (the spatial distribution of the motor on the spindle was not assessed). This is an important result essential to

interpreting the authors' FRAP data that should be incorporated in the manuscript.

5. In response to reviewer 1 comments regarding their interpretation of the FRAP experiment the authors have further revised the corresponding manuscript text, inadvertently introducing a number of inaccurate and misleading statements. The original manuscript correctly concluded, 'IFT88 depletion reduced GFP-HSET turnover on mitotic spindle microtubules'. Instead, the authors now repeatedly and incorrectly state that depletion of IFT88 impairs HSET recruitment (eg heading page 7). In Rebuttal Figure 2, the authors show that levels of HSET on the spindle are normal. A reduced FRAP recovery (t1/2 and amplitude are not clearly distinguished) would suggest an increased residence time on microtubules, not a defect in recruitment (there is no such thing as 'recruitment following photobleaching'). This must be corrected.

6. In their response to my original point 9, the authors present several valid arguments for why the effect they see could not be due to impaired cilia-dependent signaling following IFT perturbation. It would be good to incorporate these somewhere in the text.

2nd Revision - authors' response

25 February 2020

Referee #1:

Vitre and Taulet hand in a revised version of their ms on IFT proteins and HSET to promote extracentrosome clustering in mitosis.

The new version closes the few little gaps that were left in the initial submission. The authors provide additional, well-documented and quantified rescue experiments for their IFT52 knockdowns in RPE-1 and Caco2 cells. They correlate binucleation penetrance with centrosome amplification in cancer cells and add a graphical overview to document the timing of these knock-down experiments. Additional experimental data and quantifications match suggestions of reviewers 2 and 3. The authors further carefully address formal issues that were raised in my initial review, including the discussion of the presented FRAP experiments as well as a general correction of wording and grammar.

I am, therefore, in favor of publishing the manuscript in its present form in EMBO reports.

Referee #2:

The authors address my initial concerns/comments and thus i would support its publication as it is to EMBO Reports.

The addition of insets where centrioles are visible is very helpful. However, these images are very pixelated. Can the authors add better quality insets?

Re: We used optimal quality images for the insets. The large pixel size visible in our images is due to the hardware used during image acquisition: we used microscopes equipped with EMCCD camera that have large pixel size (13 μm for Andor IX888 camera and 16 μm pixel size for Andor IX897 camera) compared to most sCMOS camera that usually have smaller pixel size (6,5 μm for most sCMOS cameras). Therefore, the images acquired with those EMCCD camera show bigger pixels.

Referee #3:

The revised manuscript by Vitre et al largely addresses the points made in my original review. I am therefore in favor of publication in EMBO Reports. However, a few misstatements and inaccuracies persist or have been introduced in the revision as detailed below. I strongly recommend that the authors consider the following points in preparing their final manuscript draft:

1. In their discussion of the tracking experiment presented in Figure 4 (now clarified to be a centrosome, not centriole, tracking experiment, to my original point 3), the authors make the following statement: 'Those data indicate that HSET activity on centrosome dynamics starts before NEB and is mediated by IFT52' (page 10). I am not sure from what data the authors infer such a dependency relationship. While the phenotype the authors observe following IFT52 depletion may be stronger than that following HSET inhibition, this does not place HSET upstream (or downstream) of IFT52.

Re: We have now changed the sentences to remove the dependency relationship (page 10).

2. To my original point 4, I appreciate the authors efforts to improve the manuscript text. To me it still reads incredibly dense and occasionally confusing, but still an improvement on the original. One point the authors may still wish to consider is whether the 'extra' in extra centrosome clustering is strictly necessary. In the title it may be worth stating 'IFT proteins interact with HSET to promote clustering of supernumerary centrosomes in mitosis'. Elsewhere in the text the 'extra' can be taken as understood.

Re: According to the referee's suggestion, we removed the term "extra" when mentioning centrosome clustering. We also change in the title extra centrosome clustering with "supernumerary" centrosome clustering.

3. Addressing my original point 5, Figure EV1A now presents a quantification of centrosome amplification after PLK4 induction. What is left unclear is how centrosome number was measured and what the authors consider as abnormal, ie >4 centrin foci or >2 gamma-tubulin?

Re: We considered cells had centrosome amplification when they displayed more than two centrosomes that were defined as two centrin dots colocalizing with one gamma tubulin signal. We now clarify this in the legend of EV1A by adding the sentence: "Centrosomes were identified by the colocalization of two centrin dots with gamma-tubulin signal." We have also modified the y axis legend of figure EV1A to make this clear.

4. Rebuttal Figure 2 shows that the levels of HSET on the mitotic spindle are unaffected by inhibition with CW069 or depletion of IFT88, addressing part of my original point 6 (the spatial distribution of the motor on the spindle was not assessed). This is an important result essential to interpreting the authors' FRAP data that should be incorporated in the manuscript.

Re: We have now included rebuttal figure 2 in figure EV3 and modified the text accordingly.

5. In response to reviewer 1 comments regarding their interpretation of the FRAP experiment the authors have further revised the corresponding manuscript text, inadvertently introducing a number of inaccurate and misleading statements. The original manuscript correctly concluded, 'IFT88 depletion reduced GFP-HSET turnover on mitotic spindle microtubules'. Instead, the authors now repeatedly and incorrectly state that depletion of IFT88 impairs HSET recruitment (eg heading page 7). In Rebuttal Figure 2, the authors show that levels of HSET on the spindle are normal. A reduced FRAP recovery (t1/2 and amplitude are not clearly distinguished) would suggest an increased residence time on microtubules, not a defect in recruitment (there is no such thing as 'recruitment following photobleaching'). This must be corrected.

Re: We agree with referee 3 that recruitment is not affected as shown in rebuttal Figure 2 (now Figure EV3A). Thus, we used the term "recruitment following photobleaching", but we agree that the term is confusing. Following referee 3's advice, we now mention HSET "turnover on microtubules" in the new version of the text (page 9).

6. In their response to my original point 9, the authors present several valid arguments for why the effect they see could not be due to impaired cilia-dependent signaling following IFT perturbation. It would be good to incorporate these somewhere in the text.

Re: We have now included those arguments in the result section of our manuscript." Of note, the effect observed on centrosome clustering using DLD-1 cells and auxin inducible degradation of IFT88 cannot be due to impaired cilia-dependent signaling following IFT perturbation because DLD-1 cells do not grow cilia [42]. Moreover, cells are arrested in mitosis for two hours and acutely depleted of IFT88 in mitosis, using the AID system. In this experiment, IFT88 depletion only occurs in mitosis, where cilia are absent, therefore precluding any cilia dependent perturbation. " (page 7).

Accepted

12 March 2020

Thank you for submitting your revised manuscript. I have now looked at everything and all looks fine. Therefore I am very pleased to accept your manuscript for publication in EMBO Reports.

Congratulations on a nice work!

Corresponding Author Name: Benjamin VITRE

Manuscript Number: EMBOR-2019-49234V2